# Improving climate model coupling through a complete mesh representation: a case study with E3SM (v1) and MOAB (v5.x)

Vijay S. Mahadevan[1], Iulian Grindeanu[1], Robert Jacob[1], and Jason Sarich[1]

[1]Argonne National Laboratory, 9700 S. Cass Avenue, Lemont, IL

**Correspondence:** V. S. Mahadevan (mahadevan@anl.gov)

**Abstract.**

One of the fundamental factors contributing to the spatiotemporal inaccuracy in climate modeling is the mapping of solution field data between different discretizations and numerical grids used in the coupled component models. The typical climate computational workflow involves evaluation and serialization of the remapping weights during the pre-processing step, which is then consumed by the coupled driver infrastructure during simulation to compute field projections. Tools like Earth System Modeling Framework (ESMF) (Hill et al., 2004) and TempestRemap (Ullrich et al., 2013) offer capability to generate conservative remapping weights, while the Model Coupling Toolkit (MCT) (Larson et al., 2001) that is utilized in many production climate models exposes functionality to make use of the operators to solve the coupled problem. However, such multi-step processes present several hurdles in terms of the scientific workflow, and impedes research productivity. In order to overcome these limitations, we present a fully integrated infrastructure based on the Mesh Oriented datABase (MOAB) (Tautges et al., 2004; Mahadevan et al., 2015) library, which allows for a complete description of the numerical grids, and solution data used in each submodel. Through a scalable advancing front intersection algorithm, the supermesh of the source and target grids are computed, which is then used to assemble the high-order, conservative and monotonicity preserving remapping weights between discretization specifications. The Fortran compatible interfaces in MOAB are utilized to directly link the submodels in the Energy Exascale Earth System Model (E3SM) to enable online remapping strategies in order to simplify the coupled workflow process. We demonstrate the superior computational efficiency of the remapping algorithms in comparison with other state-of-science tools and present strong scaling results on large-scale machines for computing remapping weights between the spectral-element atmosphere and finite-volume discretizations on the polygonal ocean grids.

## 1 Introduction

Understanding Earth's climate evolution through robust and accurate modeling of the intrinsically complex, coupled ocean-atmosphere-land-ice-biosphere models requires extreme-scale computational power (Washington et al., 2008). In such coupled applications, the different component models may employ unstructured spatial meshes that are specifically generated to resolve problem-dependent solution variations, which introduces several challenges in performing a consistent solution coupling. It is known that operator decomposition and unresolved coupling errors in partitioned atmosphere and ocean model simulations (Beljaars et al., 2017), or physics and dynamics components of an atmosphere, can lead to large approximation errors that cause

severe numerical stability issues. In this context, one factor contributing to the spatiotemporal accuracy is the mapping between different discretizations of the sphere used in the components of a coupled climate model. Accurate remapping strategies in such multi-mesh problems are critical to preserve higher order resolution, but are in general computationally expensive given the disparate spatial scales across which conservative projections are calculated. Since the primal solution or auxiliary derived data defined on a donor physics component mesh (source model) needs to be transferred to its coupled dependent physics mesh (target model), robust numerical algorithms are necessary to preserve discretization accuracy during these operations (Grandy, 1999; de Boer et al., 2008), in addition to conservation and monotonicity properties in the field profile.

An important consideration is that in addition to maintaining the overall discretization accuracy of the solution during remapping, global conservation, and sometimes local element-wise conservation for critical quantities (Jiao and Heath, 2004) needs to be imposed during the workflow. Such stringent requirements on key flux fields that couple components along boundary interfaces is necessary in order to mitigate any numerical deviations in coupled climate simulations. Note that these physics meshes are usually never embedded or are not linked by any trivial linear transformations, which render existence of exact projection or interpolation operators unfeasible, even if the same continuous geometric topology is discretized in the models. Additionally, the unique domain decomposition used for each of the component physics meshes complicates the communication pattern during intra-physics transfer, since aggregation of point location requests needs to be handled efficiently in order to reduce overheads during the remapping workflow (Plimpton et al., 2004; Tautges and Caceres, 2009).

Adaptive block-structured cubed-sphere or unstructured refinement of icosahedral/polygonal meshes (Slingo et al., 2009) are often used to resolve the complex fluid dynamics behavior in atmosphere and ocean models efficiently. In such models, conservative, local flux-preserving remapping schemes are critically important (Berger, 1987) to effectively reduce multimesh errors, especially during computation of tracer advection such as water vapor or $CO_2$ (Lauritzen et al., 2010). This is also an issue in atmosphere models where physics and dynamics are computed on non-embedded grids (Dennis et al., 2012), and the improper spatial coupling between these multi-scale models could introduce numerical artifacts. Hence, the availability of different consistent and accurate remapping schemes under one flexible climate simulation framework is vital to better understand the pros and cons of the adaptive multiresolution choices (Reichler and Kim, 2008).

## 1.1 Hub-and-Spoke vs Distributed Coupling Workflow

The hub-and-spoke centralized model as shown in Fig. 1 (left) is used in the current Exascale Earth System Model (E3SM) driver, and relies on several tools and libraries that have been developed to simplify the regridding workflow within the climate community. Most of the current tools used in E3SM and the Community Earth System Model (CESM) (Hurrell et al., 2013) are included in a single package called the Common Infrastructure for Modeling the Earth (CIME), which builds on previous couplers used in CESM (Craig et al., 2005, 2012). These modeling tools approach the problem in a two-step computational process:

1. Compute the projection or remapping weights for a solution field from a source component physics to a target component physics as an offline process

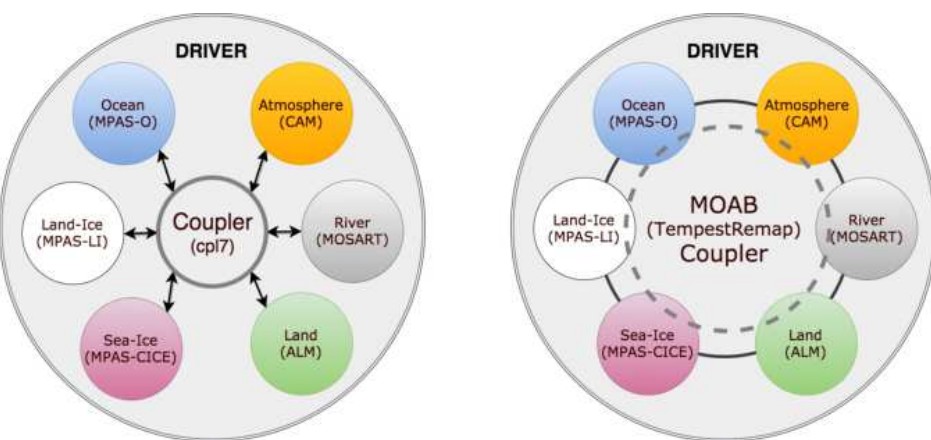

**Figure 1.** E3SM Coupled Climate Solver: (a) Current model (left), (b) Newer MOAB based coupler (right).

   2. During runtime, the CIME coupled solver loads the remapping weights from a file, and handles the partition-aware communication and weight matrix application to project coupled fields between components

The first task in this workflow is currently accomplished through a variety of standard state-of-science tools such as the Earth Science Modeling Framework (ESMF) (Hill et al., 2004), Spherical Coordinate Remapping and Interpolation Package
(SCRIP) (Jones, 1999), TempestRemap (Ullrich et al., 2013; Ullrich and Taylor, 2015). The Model Coupling Toolkit (MCT) (Larson et al., 2001; Jacob et al., 2005a) used in the CIME solver provides data structures for the second part of the workflow. Traditionally the first workflow phase is executed decoupled from the simulation driver during a pre-processing step, and hence any updates to the field discretization or the underlying mesh resolution immediately necessitates recomputation of the remapping weight generation workflow with updated inputs. This process flow also prohibits the component solvers from performing
any runtime spatial adaptivity, since the remapping weights have to be re-computed dynamically after any changes in grid positions. To overcome such deficiencies, and to accelerate the current coupling workflow, recent efforts have been undertaken to implement a fully integrated remapping weight generation process within E3SM using a scalable infrastructure provided by the topology, decomposition and data-aware Mesh Oriented datABase (MOAB) (Tautges et al., 2004; Mahadevan et al., 2015) and TempestRemap (Ullrich et al., 2013) software libraries as shown in Fig. 1 (right). Note that whether a hub-and-spoke or
distributed coupling model is used to drive the simulation, a minimal layer of driver logic is necessary to compute weighted combination of fluxes, validation metrics, and other diagnostic outputs.

The paper is organized as follows. In Section. (2), we present the necessary background and motivations to develop an online remapping workflow implementation in E3SM. Section. (3) covers details on the scalable, mesh and partition aware, conservative remapping algorithmic implementation to improve scientific productivity of the climate scientists, and to simplify
the overall computational workflow for complex problem simulations. Then, the performance of these algorithms are first evaluated in serial for various grid combinations, and the parallel scalability of the workflow is demonstrated on large-scale machines in Section. (4).

## 2 Background

Conservative remapping of nonlinearly coupled solution fields is a critical task to ensure consistency and accuracy in climate and numerical weather prediction simulations (Slingo et al., 2009). While there are various ways to compute a projection of a solution defined on a source grid $\Omega_S$ to a target grid $\Omega_T$, the requirements related to global or local conservation in the remapped solution reduces the number of potential algorithms that can be employed for such problems.

Depending on whether (global or local) conservation is important, and if higher-order, monotone interpolators are required, there are several consistent algorithmic options that can be used (de Boer et al., 2008). All of these different remapping schemes usually have one of these characteristic traits: non-conservative (**NC**), globally-conservative (**GC**) and locally-conservative (**LC**). Note that strong local-conservation prescriptions also guarantee global-conservation for the remapped fields.

1. **NC/GC**: Solution interpolation approximations

    – **NC**: (Approximate or exact) nearest neighbor interpolation

    – **NC/GC**: Radial Basis Function (RBF) (Flyer and Wright, 2007) interpolators and patch-based Least Squares reconstructions (Zienkiewicz and Zhu, 1992; Fleishman et al., 2005)

    – **GC**: Consistent Finite Element (FE) interpolation (bilinear, biquadratic, etc) with area re-normalization

2. **LC**: Mass ($L_2$) and gradient-preserving ($H_1$) projections

    – Embedded Finite Element (FE), Finite Difference (FD), and Finite Volume (FV) meshes in adaptive computations

    – Intersection-based field integrators with consistent higher-order discretization (Jones, 1999)

    – Constrained projections to ensure conservation (Berger, 1987; Aguerre et al., 2017) and monotonicity (Rančić, 1995)

Typically in climate applications, flux fields are interpolated using first-order (locally) conservative interpolation, while other scalar fields use non-conservative but higher-order interpolators (e.g. bilinear or biquadratic). For scalar solutions that do not need to be conserved, consistent FE interpolation, patch-wise reconstruction schemes (Fornberg and Piret, 2008) or even nearest neighbor interpolation (Blanco and Rai, 2014) can be performed efficiently using Kd-tree based search-and-locate point infrastructure. Vector fields like velocities or wind stresses are interpolated using these same routines by separately tackling each Cartesian-decomposed component of the field. However, conservative remapping of flux fields require computation of a supermesh (Farrell and Maddison, 2011), or a global intersection mesh that can be viewed as $\Omega_S \bigcup \Omega_T$, which is then used to compute projection weights that contain additional conservation and monotonicity constraints embedded in them.

In general, remapping implementations have three distinct steps to accomplish the projection of solution fields from a source to a target grid. First, the target points of interest are identified and located in the source grid, such that, the target cells are a subset of the covering (source) mesh. Next, an intersection between this covering (source) mesh and the target mesh is performed, in order to calculate the individual weight contribution to each target cell, without approximations to the component

field discretizations that can be defined with arbitrary-order FV or FE basis. Finally, application of the weight matrix yields the projection required to conservatively transfer the data onto the target grid.

To illustrate some key differences between some **NC** to **GC** or **LC** schemes, we show a 1-D Gaussian hill solution, projected onto a coarse grid through linear basis interpolation and weighted Least-Squares ($L_2$) minimization, as shown in Fig. 2. While the point-wise linear interpolator is computationally efficient, and second-order accurate (Fig. 2-(a)) for smooth profiles, it does not preserve the exact area under the curve. In contrast, the $L_2$ minimizer conserves the global integral area, but can exhibit spurious oscillatory modes as shown in Fig. 2-(b), when dealing with solutions with strong gradients (Gibbs phenomenon (Gottlieb and Shu, 1997)). This demonstration confirms that even for the simple 1-D example, a conservative and monotonic projector is necessary to preserve both stability and accuracy for repeated remapping operator applications, in order to accurately transfer fields between grids with very different resolutions. These requirements are magnified manyfold when dealing with real-world climate simulation data.

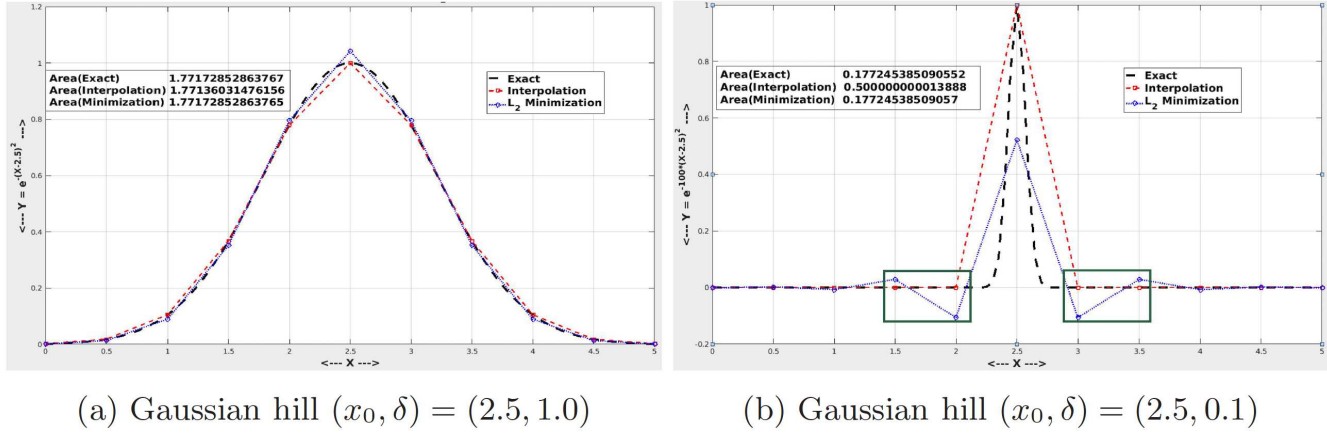

(a) Gaussian hill $(x_0, \delta) = (2.5, 1.0)$      (b) Gaussian hill $(x_0, \delta) = (2.5, 0.1)$

**Figure 2.** An illustration: comparing point interpolation vs $L_2$ minimization; impact on conservation and monotonicity properties.

While there is a delicate balance in optimizing the computational efficiency of these operations without sacrificing the numerical accuracy or consistency of the procedure, several researchers have implemented algorithms that are useful for a variety of problem domains. In the recent years, the growing interest to rigorously tackle coupled multiphysics applications has led to research efforts focused on developing new regridding algorithms. The Data Transfer Kit (DTK) (Slattery et al., 2013) from Oak Ridge National Labs was originally developed for Nuclear engineering applications, but has been extended for other problem domains through custom adaptors for meshes. DTK is more suited for non-conservative interpolation of scalar variables with either mesh-aware (using consistent discretization bases) or RBF-based meshless (point-cloud) representations (Slattery, 2016) that can be extended to model transport schemes on a sphere (Flyer and Wright, 2007). The Portage library (Herring et al., 2017) from Los Alamos National Laboratory also provides several key capabilities that are useful for geology and geophysics modeling applications including porous flow and seismology systems. Using advanced clipping algorithms to compute the intersection of axis-aligned squares/cubes against faces of a triangle/tetrahedron in 2-d and 3-d respectively,

general intersections of arbitrary convex polyhedral domains can be computed efficiently (Powell and Abel, 2015). Support for conservative solution transfer between grids and bound-preservation (to ensure monotonicity) (Certik et al., 2017) has also been recently added. While Portage does support hybrid level parallelism (MPI + OpenMP), demonstrations on large-scale machines to compute remapping weights for climate science applications has not been pursued previously. Based on the

software package documentation, support for remapping of vector fields with conservation constraints in DTK and Portage is not directly available for use in climate workflows. Additionally, unavailability of native support for projection of high-order spectral-element data on a sphere onto a target mesh restricts the use of these tools for certain component models in E3SM.

In earth science applications, the state-of-science regridding tool that is often used by many researchers is the ESMF library, and the set of utility tools that are distributed along with it (Collins et al., 2005; Dunlap et al., 2013), to simplify the traditional

offline-online computational workflow as described in Section. (1.1). ESMF is implemented in a component architecture (Zhou, 2006) and provides capabilities to generate the remapping weights for different discretization combinations on the source and target grids in serial and parallel. ESMF provides a standalone tool, ESMF_REGRIDWEIGHTGEN, to generate *offline* weights that can be consumed by climate applications such as E3SM. ESMF also exposes interfaces that enable drivers to directly invoke the remapping algorithms in order to enable the fully-online workflow as well.

Currently, the E3SM components are integrated together in a hub-and-spoke model (Fig. 1 (left)), with the inter-model communication being handled by the Model Coupling Toolkit (MCT) (Larson et al., 2001; Jacob et al., 2005a) in CIME. The MCT library consumes the offline weights generated with ESMF or similar tools, and provides the functionality to interface with models, decompose the field data, and apply the remapping weights loaded from a file during the setup phase. Hence, MCT serves to abstract the communication of data in the E3SM ecosystem. However, without the offline remapping weight

generation phase for fixed grid resolutions and model combinations, the workflow in Fig. 1 (a) is incomplete.

Similar to the CIME-MCT driver used by E3SM, OASIS3-MCT (Valcke, 2013; Craig et al., 2017) is a coupler used by many European climate models, where the interpolation weights can be generated offline through SCRIP (included as part of OASIS3-MCT). An option to call SCRIP in an online mode is also available. The OASIS team have recently parallelized SCRIP to speed up its calculation time (Valcke et al., 2018). OASIS3-MCT also supports application of global conservation

operations after interpolation, and does not require a strict hub-and-spoke coupler. Similar to the coupler in CIME, OASIS3-MCT utilizes MCT to perform both the communication of fields between components and for application of the pre-computed interpolation weights in parallel.

ESMF and SCRIP traditionally handle only cell-centered data that targets Finite Volume discretizations (FV to FV projections), with first or second order conservation constraints. Hence, generating remapping weights for atmosphere-ocean grids

with a Spectral Element (SE) source grid definition requires generation of an intermediate and spectrally equivalent, '*dual*' grid, which matches the areas of the polygons to the weight of each Gauss-Lobatto-Legendre (GLL) nodes (Mundt et al., 2016). Such procedures add more steps to the offline process and can degrade the accuracy in the remapped solution since the original spectral order is neglected (transformation from $p$-order to first order). These procedures may also introduce numerical uncertainty in the coupled solution that could produce high solution dispersion (Ullrich et al., 2016).

To calculate remapping weights directly for high-order Spectral Element grids, E3SM uses the TempestRemap C++ library (Ullrich et al., 2013). TempestRemap is a uni-process tool focused on the mathematically rigorous implementations of the remapping algorithms (Ullrich and Taylor, 2015; Ullrich et al., 2016) and provides higher order conservative and monotonicity preserving interpolators with different discretization basis such as (Finite Volume (FV), the spectrally equivalent continuous Galerkin FE with GLL basis (cGLL), and dis-continuous Galerkin FE with GLL basis (dGLL)). This library was developed as part of the effort to fill the gap in generating consistent remapping operators for non-FV discretizations without a need for intermediate dual meshes. Computation of conservative interpolators between any combination of these discretizations (FV, cGLL, dGLL) and grid definitions are supported by TempestRemap library. However, since this regridding tool can only be executed in serial, the usage of TempestRemap prior to the work presented here has been restricted primarily to generating the required mapping weights in the offline stage.

Even though ESMF and OASIS3-MCT have been used in online remapping studies, weight generation as part of a pre-processing step currently remains the preferred workflow for many production climate models. While this decoupling provides flexibility in terms of choice of remapping tools, the data management of the mapping files for different discretizations, field constraints and grids can render provenance, reproducibility and experimentation a difficult task. It also precludes the ability to handle moving or dynamically adaptive meshes in coupled simulations. However, it should be noted that the shift of the remapping computation process from a pre-processing stage in the workflow, to the simulation stage, imposes additional onus on the users to better understand the underlying component grid properties, their decompositions, the solution fields being transferred and the preferred options for computing the weights. This also raises interesting workflow modifications to ensure verification of the online weights such that consistency, conservation and dissipation of key fields are within user-specified constraints. In the implementation discussed here, the online remapping computation uses the exact same input grids, and specifications like the offline workflow, along with ability to write the weights to file, which can be used to run detailed verification studies as needed.

There are several challenges in scalably computing the regridding operators in parallel, since it is imperative to have both a mesh- and partition-aware datastructure to handle this part of the regridding workflow. A few climate models have begun to calculate weights online as part of their regular operation. The ICON GCM (Wan et al., 2013) uses YAC (Hanke et al., 2016) and FGOALS (Li et al., 2013) uses the C-Coupler (Liu et al., 2014, 2018) framework. These codes expose both offline and online remapping capabilities with parallel decomposition management similar to the ongoing effort presented in the current work for E3SM. Both of these packages provide algorithmic options to perform in-memory search and locate operations, interpolation of field data between meshes with first order conservative remapping, higher-order patch-recovery (Zienkiewicz and Zhu, 1992) and RBF schemes and the NC nearest-neighbor queries. The use of non-blocking communication for field data in these packages align closely with scalable strategies implemented in MCT (Jacob et al., 2005b). While these capabilities are used routinely in production runs for their respective models, the motivation for the work presented here is to tackle coupled high-resolution runs on next generation architectures with scalable algorithms (the high resolution E3SM coupler routinely runs on 13,000 mpi tasks), without sacrificing numerical accuracy for all discretization descriptions (FV, cGLL, dGLL) on unstructured grids.

In the E3SM workflow supported by CIME, the ESMF-regridder understands the component grid definitions, and generates the weight matrices (offline). The CIME driver loads these operators at runtime and places them in MCT datatypes, which treat them as discrete operators to compute the interpolation or projection of data on the target grids. Additional changes in conservation requirements or monotonicity of the field data cannot be imposed as a runtime or post-processing step in such a

workflow. In the current work, we present a new infrastructure with scalable algorithms implemented using the MOAB mesh library and TempestRemap package to replace the ESMF-E3SM-MCT remapper/coupler workflow. A detailed review of the algorithmic approach used in the MOAB-TempestRemap (MBTR) workflow, along with the software interfaces exposed to E3SM is presented next.

## 3   Algorithmic approach

Efficient, conservative and accurate multi-mesh solution transfer workflows (Jacob et al., 2005b; Tautges and Caceres, 2009) are a complex process. This is due to the fact that in order to ensure conservation of critical quantities in a given norm, exact cell intersections between the source and target grids have to be computed. This is complicated in a parallel setting since the domain decompositions between the source and target grids may not have any overlaps, making it a potentially all-to-all collective communication problem. Hence, efficient implementations of regridding operators need to be mesh, resolution, field

and decomposition aware in order to provide optimal performance in emerging architectures.

Fully online remapping capability within a complex ecosystem such as E3SM requires a flexible infrastructure to generate the projection weights. In order to fullfill these needs, we utilize the MOAB mesh datastructure combined with the TempestRemap libraries in order to provide an in-memory remapping layer to dynamically compute the weight matrices during the setup phase of the simulations for static source-target grid combinations. For dynamically adaptive and moving grids, the remapping

operator can be recomputed at runtime as needed. The introduction of such a software stack allows higher order conservation of fields while being able to transfer and maintain field relations in parallel, within the context of the fully decomposed mesh view. This is an improvement to the E3SM workflow where MCT is oblivious to the underlying mesh datastructure in the component models. Having a fully mesh-aware implementation with element connectivity and adjacency information, along with parallel ghosting and decomposition information also provides opportunities to implement dynamic load-balancing algorithms to gain

optimal performance on large-scale machines. Without the mesh topology, MCT is limited to performing trivial decompositions based on global ID spaces during mesh migration operations. YAC interpolator (Hanke et al., 2016) and the multidimensional Common Remapping software (CoR) in C-Coupler2 (Liu et al., 2018) provide similar capabilities to perform a parallel tree-based search for point location and interpolation through various supported numerical schemes.

MOAB is a fully distributed, compact, array-based mesh datastructure, and the local entity lists are stored in ranges along

with connectivity and ownership information, rather than explicit lists, thereby leading to a high degree of memory compression. The memory constraints per process scales well in parallel (Tautges and Caceres, 2009), and is only proportional to the number of entities in the local partition, which reduces as number of processes increases (strong scaling limit). This is similar to the Global Segment Map (GSMap) in MCT, which in contrast is stored in every processor, leading to $O(N_x)$ memory

requirements. The parallel communication infrastructure in MOAB is heavily leveraged (Tautges et al., 2012) to utilize the scalable, *crystal router algorithm* (Fox et al., 1989; Schliephake and Laure, 2015) in order to scalably communicate the covering cells to different processors. This parallel mesh infrastructure in MOAB provides the necessary algorithmic tools for optimally executing online remapping strategies, so that MCT in E3SM can be replaced with a MOAB-based coupler.

In order to illustrate the online remapping algorithm implemented with the MOAB-TempestRemap infrastructure, we define the following terms. Let $N_{c,S}$ be the component processes for source mesh, $N_{c,T}$ be the component processes for target mesh and $N_x$ be the coupler processes where the remapping operator is computed. More generally, the problem statement can be defined as: transfer a solution field $U$ defined on the domain $\Omega_S$ and processes $N_{c,S}$, to the domain $\Omega_T$ and processes $N_{c,T}$, through a centralized coupler with domain information $\Omega_S \bigcup \Omega_T$ defined on $N_x$ processes. Such a complex online remapping

workflow for projecting the field data from a source to target mesh follows the algorithm shown in Algorithm. 1.

In the following sections, the new E3SM online remapping interface implemented with a combination of the MOAB and TempestRemap libraries is explained. Details regarding the algorithmic aspects to compute conservative, high-order remapping weights in parallel, without sacrificing discretization accuracy on next generation hardware are presented.

### 3.1   Interfacing to Component Models in E3SM

Within the E3SM simulation ecosystem, there are multiple component models (atmosphere-ocean-land-ice-runoff) that are coupled to each other. While the MCT infrastructure primarily manages the global Degree-of-Freedom (DoF) partitions without a notion of the underlying mesh, the new MOAB-based coupler infrastructure provides the ability to natively interface to the component mesh, and intricately understand the field DoF data layout associated with each model. MOAB can recognize the difference between values on a cell center and values on a cell edge or corner. In the current work, the MOAB mesh database

has been used to create the relevant integration abstraction for the HOMME atmosphere model (Thomas and Loft, 2005; Taylor et al., 2007) (cubed-sphere SE grid) and the Model for Prediction Across Scales (MPAS) ocean model (Ringler et al., 2013; Petersen et al., 2015) (polygonal meshes with holes representing land and ice regions). Since details of the mesh are not available at the level of the coupler interface, additional MOAB (Fortran) calls via the `iMOAB` interface are added to HOMME and MPAS component models to describe the details of the unstructured mesh to MOAB with explicit vertex and element connectivity information, in contrast to MCT coupler that is oblivious to the underlying grid. The atmosphere-ocean coupling

requires the largest computational effort in the coupler (since they cover about 70% of the coupled domain), and hence bulk of discussions in the current work will focus on remapping and coupling between these two component models.

MOAB can handle the finite-element zoo of elements on a sphere (triangles, quadrangles, and polygons) making it an appropriate layer to store both the mesh layout (vertices, elements, connectivity, adjacencies) and the parallel decomposition for the

component models along with information on shared and ghosted entities. While having a uniform partitioning methodology across components may be advantageous for improving the efficiency of coupled climate simulations, the parallel partition of the meshes are chosen according to the requirements in individual component solvers. Fig. 3 shows examples of partitioned SE and MPAS meshes, visualized through the native MOAB plugin for VisIt (VisIt, 2005).

**Algorithm 1** MOAB-TempestRemap parallel regridding workflow

---

1: **Input**: Partitioned and distributed native component meshes on $N_{c,S}$ source and $N_{c,T}$ target processes

2: **Result**: Remapping weight matrix $W_{S \to T}$ computed for a source ($S$) and target ($T$) mesh pair on $N_x$ coupler processes

3: **Scope:** Coupler $N_x \leftarrow$ component mesh $N_{c,l}$, where $l \in [S,T]$

4: **for** each component $l \in [S,T]$ **do**

5:     – **create in-memory copy** of component unstructured mesh and data using MOAB interfaces (Section. 3.1)

6:     – **migrate** MOAB component mesh to coupler; repartition from $N_{c,l} \to N_x$ (Section. 3.2)

7: **end for**

8: **Scope:** Compute pair-wise intersection mesh on coupler processes $N_x$

9: **for** each mesh pair to be regridded: $\Omega_S$ and $\Omega_T$ in $N_x$ **do**

10:     **Ensure:** {local source mesh fully covers target mesh}

11:     **if** $(\Omega_T - \Omega_T \cap \Omega_S) \neq 0$ **then**

12:         collectively gather coverage mesh $\Omega_{Sc}$ on $N_x \mid (\Omega_T - \Omega_T \cap \Omega_{Sc}) = 0$ (Section. 3.4.1)

13:     **end if**

14:     – **store communication graph** to send/receive between $N_{c,l}$ and $N_x$

15:     – **compute** $\Omega_{ST} = \Omega_{Sc} \cap \Omega_T$ through an *advancing-front algorithm* (Löhner and Parikh, 1988; Gander and Japhet, 2009) (Section. 3.3.1)

16:     – **evaluate source/target element mapping** for $e_i \in \Omega_{ST}$

17:     – **exchange ghost cell information** for $\Omega_{ST}$

18: **end for**

19: **Scope:** Integrate over $\Omega_{ST}$ to compute remapping weights

20: **for** each intersection polygon element $e_i \in \Omega_{ST}$ **do**

21:     – **Tessellate** $e_i$ into triangular elements with reproducible ordering

22:     – **Compute projection integral** with consistent Triangular quadrature rules

23:     – **Determine row/col DoF coupling** through $e_i$ parent association to $\Omega_S/\Omega_T$

24:     – **Assemble local matrix weights** such that $W_{S \to T} = \sum_1^{N_x} w_{ij}$, where $w_{ij}$ represents the coupling between local target DoF (row $i$) and source DoF (col $j$) in projection operator (Section. 3.5)

25: **end for**

---

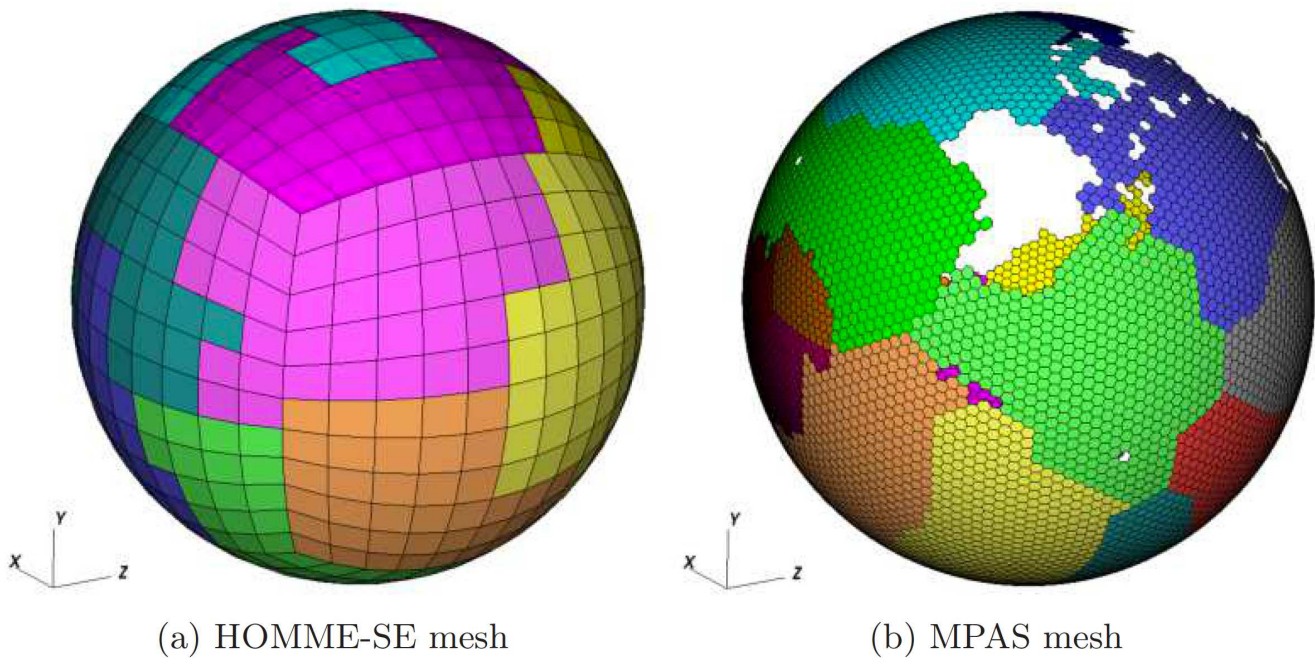

(a) HOMME-SE mesh         (b) MPAS mesh

**Figure 3.** MOAB representation of partitioned component meshes.

The coupled field data that is to be remapped from the source grid to the target grid also needs to be serialized as part of the MOAB mesh database in terms of an internally contiguous, MOAB data storage structure named a 'Tag' (Tautges et al., 2004). For E3SM, we use element-based tags to store the partitioned field data that is required to be remapped between components. Typically, the number of DoF per element ($nDoF_e$) is determined based on the underlying discretization; $nDoF_e = p^2$ values

in HOMME where $p$ is the order of SE discretization, and $nDoF_e = 1$ for the FV discretization in MPAS ocean. With this complete description of the mesh and associated data for each component model, MOAB contains the necessary information to proceed with the remapping workflow.

### 3.2 Migration of Component Mesh to Coupler

E3SM's driver supports multiple modes of partitioning the various components in the global processor space. This is usually

fine tuned based on the estimated computational load in each physics, according to the problem case definition. A sample process-execution (PE) layout for a E3SM run on 9000 processes with ATM on 5400 and OCN on 3600 processes is shown in Fig. 4. In the case shown in the schematic, $N_{c,ATM} = 5400$, $N_{c,OCN} = 3600$ and $N_x = 4800$. In such a PE layout, the atmosphere component mesh from HOMME, distributed on $N_{c,ATM}$ (5400) processes needs to be migrated and redistributed on $N_x$ (4800) processes. Similarly, from $N_{c,OCN}$ (3600) to $N_x$ (4800) processes for the MPAS ocean mesh. In the hub-and-

spoke coupling model as shown in Fig. 1, the remapping computation is performed only in the coupler processors. Hence, inference of a communication pattern becomes necessary to ensure scalable data transfers between the components and the

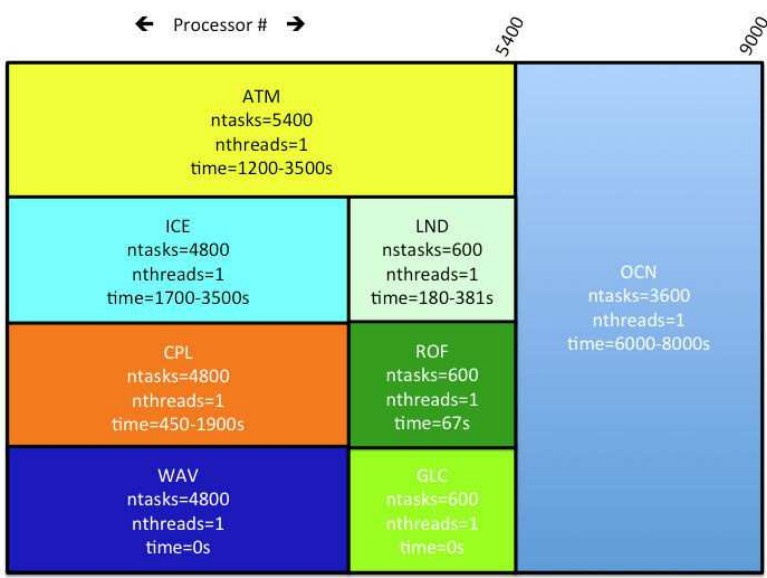

**Figure 4.** Example E3SM process execution layout for a problem case

coupler. In the existing implementation, MCT handles such communication, which is being replaced by point-to-point communication kernels in MOAB to transfer mesh and data between different components or component-coupler PEs. Note that in a distributed coupler, source and target components can communicate directly, without any intermediate transfers (through the coupler). Under the unified infrastructure provided by MOAB, minimal changes are required to enable either the hub-and-

spoke or the distributed coupler for E3SM runs, which offers opportunities to minimize time to solution without any changes in spatial coupling behavior.

For illustration, let $N_c$ be the number of component process elements, and $N_x$ be the number of coupler process elements. In order to migrate the mesh and associated data from $N_c$ to $N_x$, we first compute a trivial partition of elements that map directly in the partition space, the same partitioning as used in the CIME-MCT coupler. In MOAB, we have exposed parallel graph and

geometric repartitioning schemes through interfaces to Zoltan (Devine et al., 2002) or ParMetis (Karypis et al., 1997), in order to evaluate optimized migration patterns to minimize the volume of data communicated between component and coupler. We intend to analyze the impact of different migration schemes on the scalability of the remapping operation in Section. (4). These optimizations have the potential to minimize data movement in the MOAB-based remapper, and to make it a competitive data broker to replace the current MCT (Jacob et al., 2005a) coupler in E3SM.

We show an example of a decomposed ocean mesh (polygonal MPAS mesh) that is replicated in a E3SM problem case run on two processes in Fig. 5. Fig. 5-(a) is the original decomposed mesh on 2 processes $\in N_c$, while Fig. 5-(b) and Fig. 5-(c) show the impact of migrating a mesh from 2 $N_c$ processes to 4 processes $\in N_x$ with a trivial linear partitioner and a Zoltan based geometric online partitioner. The decomposition in Fig. 5-(b) shows that the element ID based linear partitioner can produce

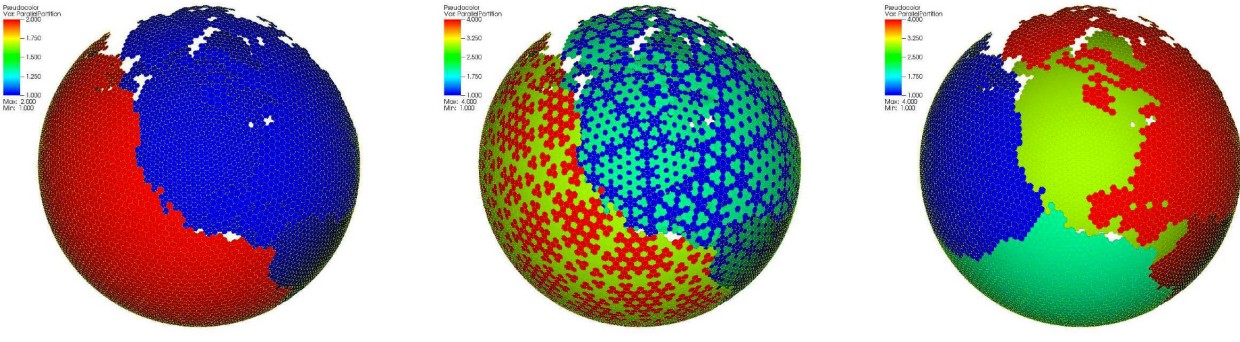

(a) Component mesh on 2 tasks     (b) Migrated mesh on 4 tasks (Trivial partitioner)     (c) Migrated mesh on 4 tasks (Zoltan partitioner)

**Figure 5.** Migration strategies to repartition from $N_c \rightarrow N_x$

bad data locality, which may require large number of nearest neighbor communications when computing a source coverage mesh. The resulting communication pattern can also make the migration, and coverage computation process non-scalable on larger core counts. In contrast, in Fig. 5-(c), the Zoltan partitioners produce much better load balanced decompositions with Hypergraph (PHG), Recursive Coordinate Bisection (RCB) or Recursive Inertial Bisection (RIB) algorithms to reduce communication overheads in the remapping workflow. In order to better understand the impact of online decomposition strategies on the overall remapping operation, we need to better understand the impact of the repartitioner on two communication-heavy steps.

1. Mesh migration from component to coupler involving communication between $N_{c,s/t}$ and $N_x$,

2. Computing the coverage mesh requiring gather/scatter of source mesh elements to cover local target elements.

In a hub-and-spoke model with online remapping, the best coupler strategy will require a simultaneous partition optimization for all grids such that mesh migration includes constraints on geometric coordinates of component pairs. While such extensions can be implemented within the infrastructure presented here, the performance discussions in Section 4 will only focus on the trivial and Zoltan-based partitioners. It is also worth noting that in a distributed coupler, pair-wise migration optimizations can be performed seamlessly using a master(*target*)-slave(*source*) strategy to maximize partition overlaps.

### 3.3 Computing the Regridding Operator

Standard approaches to compute the intersection of two convex polygonal meshes involve the creation of a Kd-tree (Hunt et al., 2006) or BVH-tree datastructure (Ize et al., 2007) to enable fast element location of relevant target points. In general, each target point of interest is located on the source mesh by querying the tree datastructure, and the corresponding (source) element is then marked as a contributor to the remapping weight computation of the target DoF. This process is repeated to form a list

of source elements that interact directly according to the consistent discretization basis. TempestRemap, ESMF and YAC use variations of this search-and-clip strategy tailored to their underlying mesh representations.

### 3.3.1 Advancing Front Intersection – A Linear Complexity Algorithm

The intersection algorithm used in this paper follows the ideas from (Löhner and Parikh, 1988; Gander and Japhet, 2013), in which two meshes are covering the same domain. At the core is an advancing front method that aims to traverse through the source and target meshes to compute a union (super) mesh. First, two convex cells from the source coverage mesh and the target meshes that intersect are identified by using an adaptive Kd-tree search tree datastructure. This process also includes determination of the seed (the starting cell) for the advancing front in each of the partitions independently. Advancing in both meshes using face adjacency information, incrementally all possible intersections are computed (Březina and Exner, 2017) accurately to a user defined tolerance (default $= 1e - 15$) in linear time.

While the advancing front algorithm is not restricted to convex cells, the intersection computation is simpler if they are strictly convex. If concave polygons exist in the initial source or target meshes, they are recursively decomposed into simpler convex polygons, by splitting along interior diagonals. Note that the intersection between two convex polygons results in a strictly convex polygon. Hence, the underlying intersection algorithm remains robust to resolve even arbitrary non-convex meshes covering the same domain space.

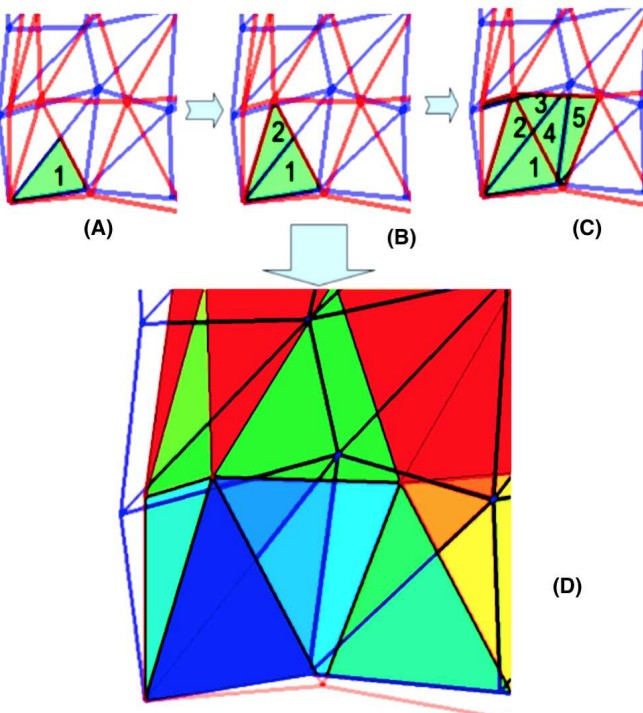

**Figure 6.** Illustration of the advancing front intersection algorithm.

Fig. 6 illustrates how the algorithm advances. In each local partition of the coupler PEs, a pair of source (blue) and target (red) cells that intersect is found Fig. 6 (A). Using face adjacency queries for the source mesh, as shown in Fig. 6 (B), all source cells that intersects the current target cell are found. New possible pairs between cells adjacent to the current target cell and other source cells are added to a local queue. After the current target cell is resolved, a pair from the queue is considered next. Fig. 6 (C) shows the resolving of a second target cell , which is intersecting here with 3 source cells. If both meshes are contiguous, this algorithm guarantees to compute all possible intersections between source and target cells. Fig. 6 (D) shows the colormap representation of the progression in which the intersection polygons were found, from blue (low count) towards red . This advance/progression is also illustrated through videos in both the serial (Mahadevan et al., 2018a) and parallel (Mahadevan et al., 2018b) contexts on partitioned meshes.

This flooding-like advancing front needs a stable and robust methodology of intersecting edges/segments in two cells that belong to different meshes. Any pair of segments that intersect can appear in four different pairs of cells. A list of intersection points is maintained on each target edge, so that the intersection points are unique. Also, a geometric tolerance is used to merge intersection points that are close to each other, or if they are proximal to the original vertices in both meshes. Decisions regarding whether points are inside, outside or at the boundary of a convex enclosure are handled separately. If necessary, more robust techniques such as adaptive precision arithmetic procedures used in *Triangle* (Shewchuk, 1996), can be employed to resolve the fronts more accurately. Note that the advancing front strategy can be employed for meshes with topological holes (e.g. ocean meshes, in which the continents are excluded) without any further modifications by using a new pair for each disconnected region in the target mesh.

**Note on Gnomonic Projection for Spherical Geometry**

Meshes that appear in climate applications are often on a sphere. Cell edges are considered to be great circle arcs. A simple gnomonic projection is used to project the edges on one of the six planes parallel to the coordinate axis, and tangent to the sphere (Ullrich et al., 2013). With this projection, all curvilinear cells on the sphere are transformed to linear polygons on a gnomonic plane, which simplifies the computation of intersection between multiple grids. Once the intersection points and cells are computed on the gnomonic plane, these are projected back on to the original spherical domain without approximations. This is possible due to the fact that intersection can be computed to machine precision as the edges become straight lines in a gnomonic plane (projected from great circle arcs on a sphere). If curves on a sphere are not great circle arcs (splines, for example), the intersection between those curves have to be computed using some nonlinear iterative procedures such as Newton Raphson (depending on the representation of the curves).

**3.4 Parallel Implementation Considerations**

Existing Infrastructure from MOAB (Tautges et al., 2004) was used to extend the advancing front algorithm in parallel. The expensive intersection computation can be carried out independently, in parallel, once we redistribute the source mesh to envelope the target mesh areas fully, in a step we refer to as 'source coverage mesh' computation.

### 3.4.1 Computation of a Source Coverage Mesh

We select the target mesh as the driver for redistribution of the source mesh. On each task, we first compute the bounding box of the local target mesh. This information is then gathered and communicated to all coupler PEs, and used for redistributing the local source mesh. Cells that intersect the bounding boxes of other processors are sent to the corresponding owner task using the aggregating *crystal router* algorithm that is particularly efficient in performing all-to-all strategies with $O(log(N_x))$ complexity. This graph is computed once during the setup phase to establish point-to-point communication patterns, which is then used to pack and send/receive mesh elements or data at runtime.

This workflow guarantees that the target mesh on each processor is completely enveloped by the covering mesh repartitioned from its original source mesh decomposition, as shown in Fig. 7. In other words, the covering mesh fully encompasses and bounds the target mesh in each task. It is important to note that some source coverage cells might be sent to multiple processors during this step, depending on the target mesh resolution and decomposition.

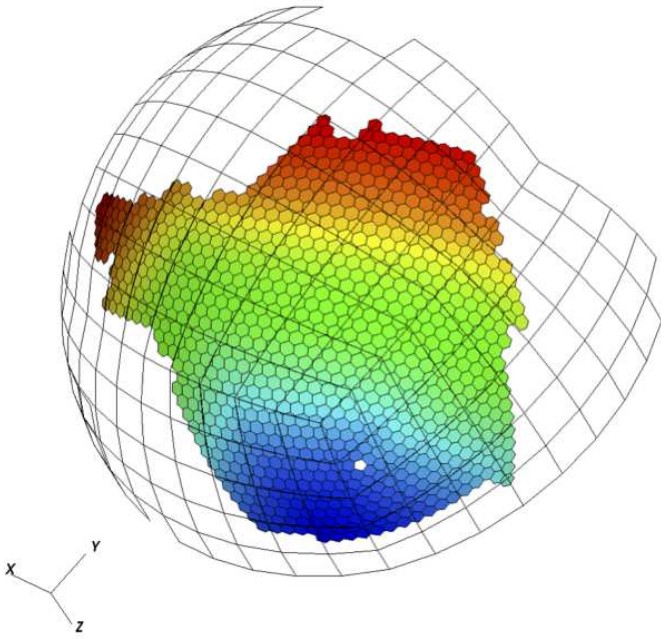

**Figure 7.** Source coverage mesh fully covers local target mesh; local intersection proceeds between the source atmosphere (Quadrangle) and the target ocean (Polygonal) grids.

Once the relevant covering mesh is accumulated locally on each process, the intersection computation can be carried out in parallel, completely independently, using the advancing front algorithm (Section. (3.3.1)). After computation of the local intersection polygons, the vertices on the shared edges between processes are communicated to avoid duplication. In order to ensure consistent local conservation constraints in the weight matrix in the parallel setting, there might be a need for additional communication of ghost intersection elements to nearest neighbors. This extra communication step is only required for

computing interpolators for flux variables, and can generally be avoided when transferring scalar fields with non-conservative bilinear or higher-order interpolations. Note that this ghost exchange on the intersection mesh only requires nearest-neighbor communications within the coupler PEs, since the communication graph has been established a-*priori*.

The parallel advancing front algorithm presented here to globally compute the intersection supermesh can be extended
to expose finer grained parallelism using hybrid-threaded (OpenMP) programming or a task-based execution model, where each task handles a unique front in the computation queue. Such task or hybrid threaded parallelism can be employed in combination with the MPI-based mesh decompositions. Using local partitions computed with Metis and through standard coloring approaches, each thread or task can then proceed to compute the intersection elements until the front collides with another, and until all the overlap elements have been computed in each process. Such a parallel hybrid algorithm has the
potential to scale well even on heterogeneous architectures and provides options to improve the computational throughput of the regridding process (Löhner, 2014).

### 3.5 Computation of Remapping operator with TempestRemap

For illustration, consider a scalar field $U$ discretized with standard Galerkin FEM on source $\Omega_1$ and target $\Omega_2$ meshes with different resolutions. The projection of the scalar field on the target grid is in general given as follows.

$$U_2(\mathbf{\Omega_2}) = \mathbf{\Pi}_1^2 U_1(\mathbf{\Omega_1}) \tag{1}$$

where, $\mathbf{\Pi}_1^2$ is the discrete solution interpolator of $U$ defined on $\mathbf{\Omega_1}$ to $\mathbf{\Omega_2}$. This interpolator $\mathbf{\Pi}_1^2$ in Eq. (1) is often referred to as the remapping operator, which is pre-computed in the coupled climate workflows using ESMF and TempestRemap. For embedded meshes, the remapping operator can be calculated exactly as a restriction or prolongation from the source to target grid. However, for general unstructured meshes and in cases where the source and target meshes are topologically different,
the numerical integration to assemble $\mathbf{\Pi}_1^2$ needs to be carried out on the supermesh (Ullrich and Taylor, 2015). Since a unique source and target parent element exists for every intersection element belonging to the supermesh $\mathbf{\Omega_1} \bigcup \mathbf{\Omega_2}$, $\mathbf{\Pi}_1^2$ is assembled as the sum of local mass matrix contributions on the intersection elements, by using the consistent discretization basis for the source and target field descriptions (Ullrich et al., 2016). The intersection mesh typically contains arbitrary convex polygons and hence subsequent triangulation may be necessary before evaluating the integration. This global linear operator directly
couples source and target DoFs based on the participating intersection element parents (Ullrich et al., 2009).

MOAB supports point-wise FEM interpolation (bilinear and higher-order spectral) with local or global subset normalization (Tautges and Caceres, 2009), in addition to a conservative first-order remapping scheme. But higher order conservative monotone weight computations are currently unsupported natively. To fill this gap for climate applications, and to leverage existing developments in rigorous numerical algorithms to compute the conservative weights, interfaces to TempestRemap in MOAB
were added to scalably compute the remap operator in parallel, without sacrificing field discretization accuracy. The MOAB interface to the E3SM component models provides access to the underlying type and order of field discretization, along with the global partitioning for the DoF numbering. Hence the projection or the weight matrix can be assembled in parallel by traversing

through the intersection elements, and associating the appropriate source and target DoF parent to columns and rows respectively. The MOAB implementation uses a sparse matrix representation using the Eigen3 library (Guennebaud et al., 2010) to store the local weight matrix. Except for the particular case of projection onto a target grid with cGLL description, the matrix rows do not share any contributions from the same source DoFs. This implies that for FV and dGLL target field descriptions, the application of the weight matrix does not require global collective operations and sparse matrix-vector (SpMV) applications scale ideally (still memory bandwidth limited). In the cGLL case, we perform a reduction of the parallel vector along the shared DoFs to accumulate contributions exactly. However, it is non-trivial to ensure full bit-for-bit (BFB) reproducibility during such reductions and currently, the MBTR workflow does not support exact reproducibility. Requirements for rigorous bitwise reproduction for online remapping needs careful implementation to enforce that the advancing-front intersection and weight matrix is computed in the exact same global element order, in addition to ensuring that the parallel SpMV products are reduced identically, independent of the parallel mesh decompositions.

It is also possible to use the transpose of the remapping operator computed between a particular source and target component combination, to project the solution back to the original source grid. Such an operation has the advantage of preserving the consistency and conservation metrics originally imposed in finding the remapping operator and reduces computation cost by avoiding recomputation of the weight matrix for the new directional pair. For example, when computing the remap operator between atmosphere and ocean models (with holes), it is advantageous to use the atmosphere model as the source grid, since the advancing front seed computation may require multiple trials if the initial front begins within a hole in the source mesh. Given that the seed or the initial cell determination on the target mesh is chosen at random, the corresponding intersecting cell on the source mesh found through a linear search could be contained within a hole in the source mesh. In such a case, a new target cell is then chosen and the source cell search is repeated. Hence multiple trials may be required for the advancing front algorithm to start propagating, depending on the mesh topology and decomposition. Note that the linear search in the source mesh can easily be replaced with a Kd-tree datastructure to provide better computational complexity for cases where both source and target meshes have many holes. Additionally, such transpose vector applications can also make the global coupling symmetric, which may have favorable implications when pursuing implicit temporal integration schemes.

### 3.6   Note on MBTR Remapper Implementation

The remapping algorithms presented in the previous section are exposed through a combination of implementations in MOAB and TempestRemap libraries. Since both the libraries are written in C++, direct inheritance of key datastructures such as the GridElements (mesh) and OfflineMap (projection weights) are available to minimize data movement between the libraries. Additionally, Fortran codes such as E3SM can invoke computations of the intersection mesh and the remapping weights through specialized language-agnostic interfaces in MOAB: iMOAB (Mahadevan et al., 2015). These interfaces offer the flexibility to query, manipulate and transfer the mesh between groups of processes that represent the component and coupler processing elements.

Using the iMOAB interfaces, the E3SM coupler can coordinate the online remapping workflow during the setup phase of the simulation, and compute the projection operators for component and scalar or vector coupled field combinations. For each

pair of coupled components, the following sequence of steps are then executed to consistently compute the remapping operator and transfer the solution fields in parallel.

1. `iMOAB_SendMesh` and `iMOAB_ReceiveMesh`: Send the component mesh (defined on $N_{c,l}$ processes), and receive the complete unstructured mesh copy in the coupler processes ($N_x$). This mesh migration undergoes an online mesh repartition either through a trivial decomposition scheme or with advanced Zoltan algorithms (geometric or graph partitioners)

2. `iMOAB_ComputeMeshIntersectionOnSphere`: The advancing front intersection scheme is invoked to compute the overlap mesh in the coupler processes

3. `iMOAB_CoverageGraph`: Update the parallel communication graph based on the (source) coverage mesh association in each process

4. `iMOAB_ComputeScalarProjectionWeights`: The remapping weight operator is computed and assembled with discretization-specific (FV, SE) calls to TempestRemap, and stored in Eigen3 SparseMatrix object

Once the remapping operator is serialized in-memory for each coupled scalar and flux fields, this operator is then used at every timestep to compute the actual projection of the data.

1. `iMOAB_SendElementTag` and `iMOAB_ReceiveElementTag`: Using the coverage graph computed previously, direct one-to-one communication of the field data is enabled between $N_{c,l}$ and $N_x$, before and after application of the weight operator

2. `iMOAB_ApplyScalarProjectionWeights`: In order to compute the field interpolation or projection from the source component to the target component, a matvec product of the weight matrix and the field vector defined on the source grid is performed. The source field vector is received from source processes $N_{c,s}$ and after weight application, the target field vector is sent to target processes $N_{c,l}$

Additionally, to facilitate offline generation of projection weights, a MOAB based parallel tool `mbtempest` has been written in C++, similar to ESMF and TempestRemap (serial) standalone tools. `mptempst` can load the source and target meshes from files, in parallel, and compute the intersection and remapping weights through TempestRemap. The weights can then be written back to a SCRIP-compatible file format, for any of the supported field discretization combinations in source and destination components. Added capability to apply the weight matrix onto the source solution field vectors, and native visualization plugins in VisIt for MOAB, simplify the verification of conservation and monotonicity for complex remapping workflows. This workflow allows users to validate the underlying assumptions for remapping solution fields across unstructured grids, and can be executed in both a serial and parallel setting.

## 4   Results

Evaluating the performance of the in-memory, MOAB-TempestRemap (MBTR) remapping infrastructure requires recursive profiling and optimization to ensure scalability for large-scale simulations. In order to showcase the advantage of using the mesh-aware MOAB datastructure as the MCT coupler replacement, we need to understand the per task performance of the regridder in addition to the parallel point locator scalability, and overall time for remapping weight computation. Note that except for the weight application for each solution field from a source grid to a target grid, the in-memory copy of the component meshes, migration to coupler PEs, computation of intersection elements and remapping weights are done only once during the setup phase in E3SM, per coupled component model pair.

### 4.1   Serial Performance

We compare the total cost for computing the supermesh and the remapping weights for several source and target grid combinations through three different methods to determine the serial computational complexity.

1. ESMF: Kd-tree based regridder and weight generation for first/second order FV→FV conservative remapping

2. TempestRemap: Kd-tree based supermesh generation and conservative, monotonic, high-order remap operator for FV→FV, SE→FV, SE→SE projection

3. MBTempest: Advancing front intersection with MOAB and conservative weight generation with TempestRemap interfaces

Fig. 8 shows the serial performance of the remappers for computing the conservative interpolator from Cubed-Sphere (CS) grids to polygonal MPAS grids of different resolutions for a FV→FV field transfer. This total time includes the computation of intersection mesh or supermesh, in addition to the remapping weights with field conservation specifications. These serial runs were executed on a machine with 8x Intel Xeon(R) CPU E7-4820 @ 2.00GHz (total of 64 cores) and 1.47 TB of RAM. As the source grid resolution increases, the advancing front intersection with linear complexity outperforms the Kd-tree intersection algorithms used by TempestRemap and ESMF. The time spent in the remapping task, including the overlap mesh generation, provides an overall metric on the single task performance when memory bandwidth or communication concerns do not dominate in a parallel run. In this comparison with three remapping software libraries, the total computational time in the fine resolution limit as $\frac{nele(source)}{nele(target)} \approx 1$ consistently increases (going diagonally from left to right in Fig. 8). We note that the serial version of TempestRemap is comparable to ESMF and can even provide better timings on the highly refined cases, while the MBTempest remapper consistently outperforms both the tools, with a 2x speedup on average. The relatively better performance in MBTempest is accomplished through the linear complexity advancing front algorithm, which further offers avenues to incorporate finer grain task or thread level parallelism to accelerate the on-node performance on multicore and GPGPU architectures.

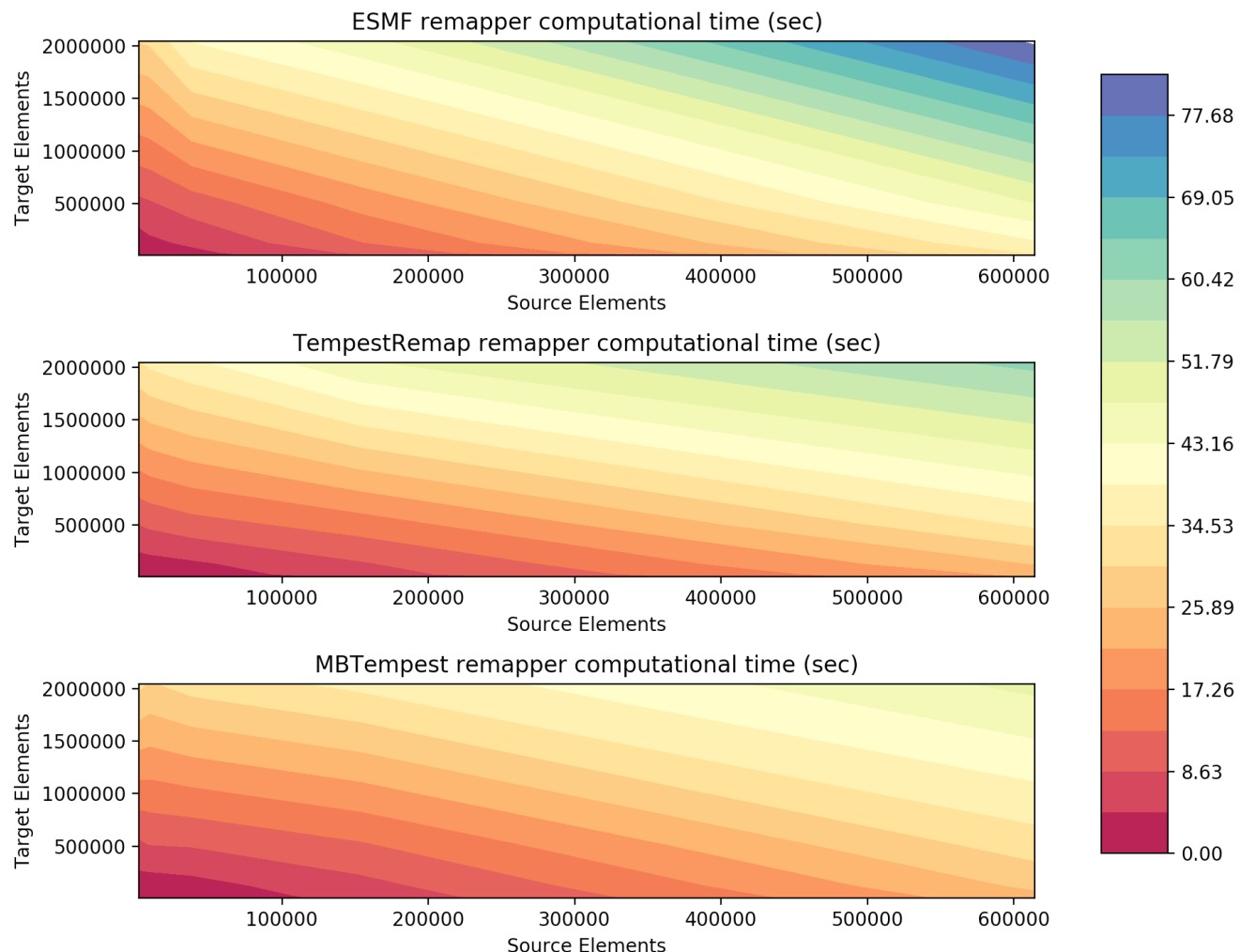

**Figure 8.** Comparison of serial regridding computation (supermesh and projection weight generation) between ESMF, TempestRemap, and MBTempest

## 4.2 Scalability of the MOAB Kd-tree Point Locator

In addition to being able to compute the supermesh between $\Omega_S$ and $\Omega_T$, MOAB also offers datastructures to query source elements containing points that correspond to the target DoFs locations. This operation is critical in evaluating bilinear and biquadratic interpolator approximations for scalar variables when conservative projection is not required by the underlying coupled model. The solution interpolation for the multi-mesh case involves two distinct phases.

1. Setup phase: Use Kd-tree to build the search datastructure to locate points corresponding to vertices in the target mesh on the source mesh

2. Run phase: Use the elements containing the located points to compute consistent interpolation onto target mesh vertices

Studies were performed to evaluate the strong and weak scalability of the parallel Kd-tree point search implementation in MOAB. The scalability results were generated with the CIAN2 coupling mini-app (Morozov and Peterka, 2016), which links to MOAB to handle traversal of the unstructured grids and transfer of solution fields between the grids. For this case, a series of hexahedral and tetrahedral meshes were used to interpolate an analytical solution. By changing the basis interpolation order, and mesh resolutions, the convergence of the interpolator was verified to provide theoretical accuracy orders of convergence in the asymptotic fine limit.

The performance tests were executed on the IBM BlueGene/Q Mira at 16 MPI ranks per node, with 2GB RAM per MPI rank, at up to 500K MPI processes. The strong scaling results and error convergence were computed with a grid size of $1024^3$. The solution interpolation on varying mesh resolutions were performed by projecting an analytical solution from a Tetrahedral→Hexahedral→Tetrahedral grid, with total number of points/rank varied between [2K, 32K] in the study. Note that the total DoFs in this study is much larger than typical climate production runs, and hence we use these experiments to showcase the strong scaling of the bilinear interpolation operation at the high-res limit.

First, the root-mean-square (RMS) error was measured in the bilinearly interpolated solution against the analytical solution, and plotted for different source and target mesh resolutions. Fig. 9-(a) demonstrates that the error convergence of the interpolants match the expected theoretical second order rates, and that the error constant is proportional to ratio of source to target mesh resolution. Next, the Fig. 9-(b) shows the strong scaling efficiency of around 50% is achieved on a maximum of 512K cores (66% of Mira). We note that the computational complexity of the Kd-tree data structure scales as $O(nlog(n))$ asymptotically, and the point location phase during initial search setup dominates the total cost on higher core counts. This is evident in the timing breakdown for each phase shown in Fig. 9-(c). Since the point location is performed only once during simulation startup, while the interpolation is performed multiple times per timestep during the run, we expect the total cost of the projection for scalar variables to be amortized over transient climate simulations with fixed grids. Further investigations with optimal BVH-tree (Larsen et al., 1999) or R-tree implementations for these interpolation cases could help reduce the overall cost.

The full 3-D point location and interpolation operations provided by MOAB are comparable to the implementation in Common Remapping component used in the C-Coupler (Liu et al., 2013) and provide relatively much stronger scalability on larger core counts (Liu et al., 2014) for the remapping operation. Such higher-order interpolators for multicomponent physics variables can provide better performance in atmospheric chemistry calculations. Additionally, as component mesh resolutions are increased to sub-Km regimes, the expectations from remapping libraries such as MOAB to provide scalable search and location of points becomes important. Currently, only the NC bilinear or biquadratic interpolation of scalar fields with subset normalization (Tautges and Caceres, 2009) is supported directly in MOAB (via Kd-tree point location and interpolation), and advancing front intersection algorithm does not make use of these data-structures. In contrast, TempestRemap and ESMF use a Kd-tree search to not only compute the location of points, but also to evaluate the supermesh $\Omega_S \bigcup \Omega_T$, and hence the computational complexity for the intersection mesh determination scales as $O(nlog(n))$, in contrast to the linear complexity ($O(n)$) of the advancing front intersection algorithm implemented in MOAB.

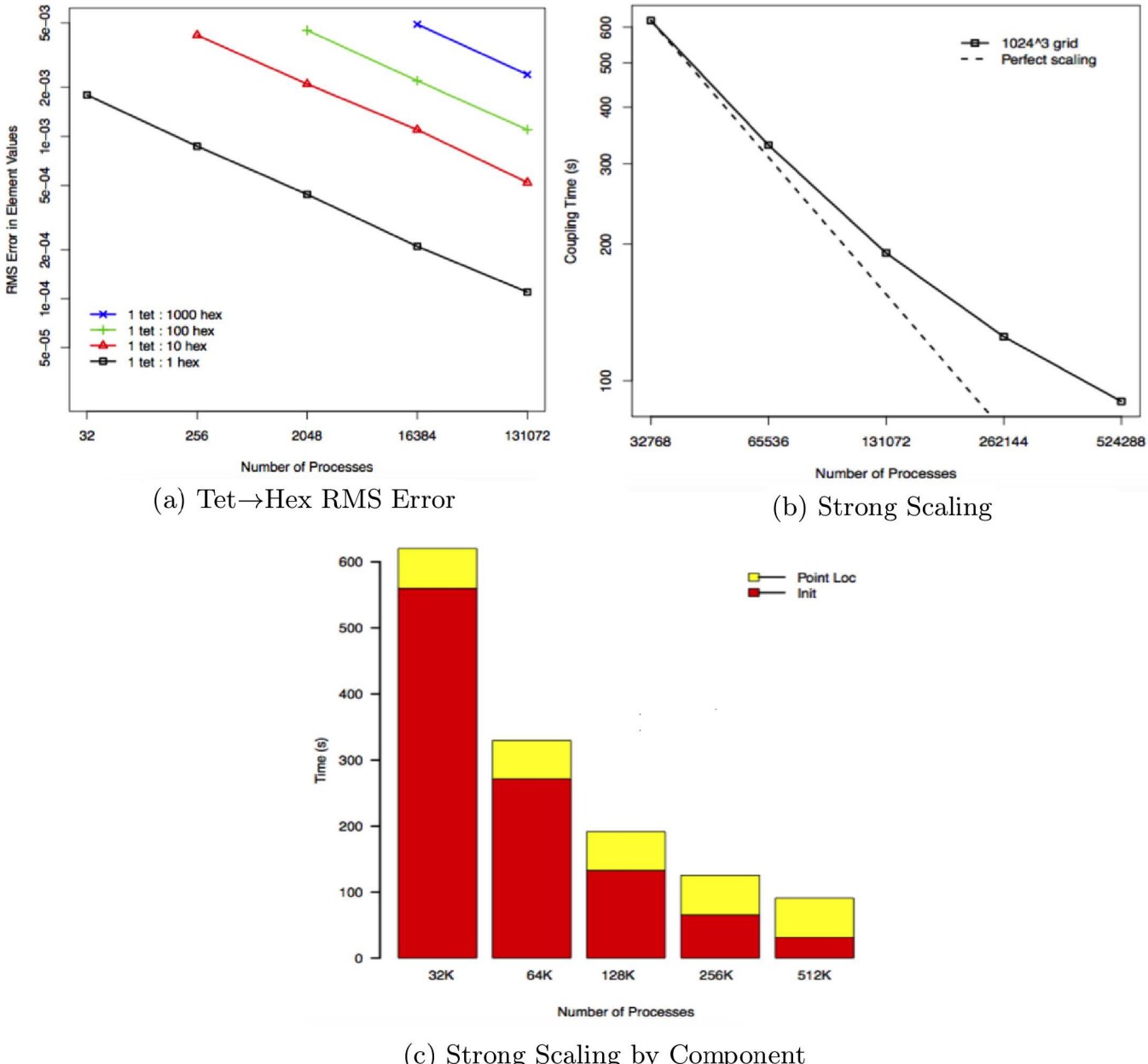

(a) Tet→Hex RMS Error

(b) Strong Scaling

(c) Strong Scaling by Component

**Figure 9.** MOAB 3-d Kd-tree Point Location: Strong scaling on Mira (BG/Q)

## 4.3 The Parallel MBTR Remapping Algorithm

The MBTR online weight generation workflow within E3SM was employed to verify and test the projection of real simulation data generated during the coupled atmosphere-ocean model runs. A choice was made to use the model-computed temperature

on the lowest level of the atmosphere, since the heat fluxes that nonlinearly couples the atmosphere and ocean models are directly proportional to this interface temperature field. By convention, the fluxes are computed on the ocean mesh, and hence the atmosphere temperature must be interpolated onto MPAS polygonal mesh. We use this scenario as a test case for demonstrating the strong scalability results in this section.

5    The atmosphere run with approximately 4 degree grid size and 11 elements per edge on a cubed-sphere (NE11) in E3SM, and the projection of its lowest level temperature onto two different MPAS meshes (with approximate grid size of 240km) are shown in Fig. 10. The conservative field projection from SE→FV on a mesh with holes corresponding to land regions is given in Fig. 10-(b), where the continents are shown in transparent shading. To contrast, we also present the remapped field on an MPAS mesh without holes (Fig. 10-(c)) to show the differences in the remapped solutions as a function of mesh topology.

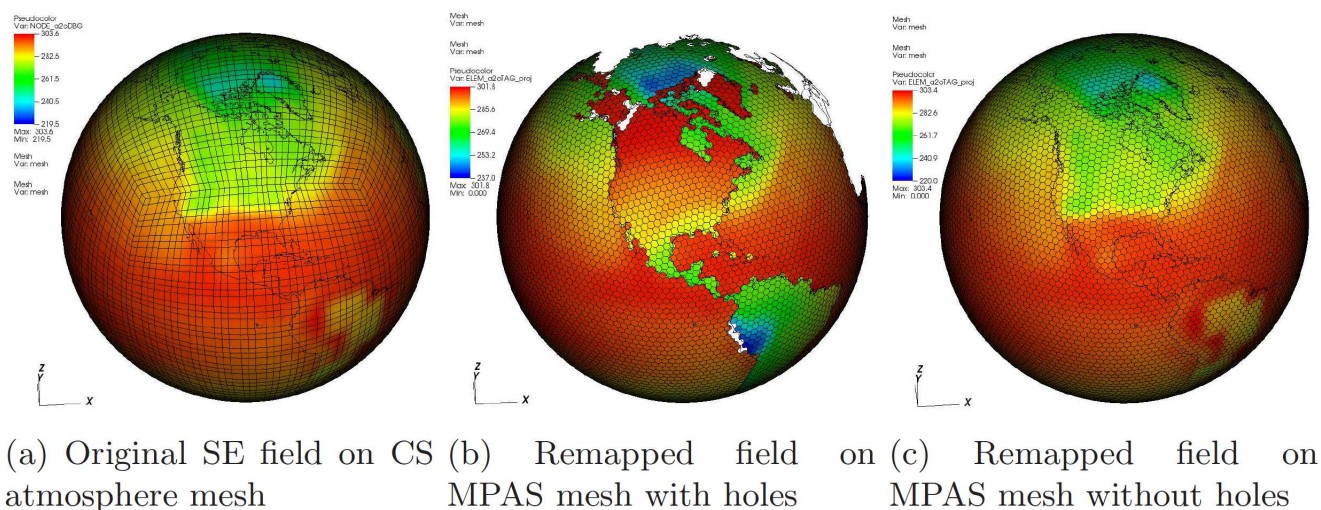

(a) Original SE field on CS atmosphere mesh

(b) Remapped field on MPAS mesh with holes

(c) Remapped field on MPAS mesh without holes

**Figure 10.** Projection of the NE11 SE bottom atmospheric temperature field onto the MPAS ocean grid

10    **4.3.1    Scaling Comparison of Conservative Remappers (FV→FV)**

The strong scaling studies for computation of remapping weights to project a FV solution field between CS grids of varying resolutions was performed on the Blues large-scale cluster (with 16 Sandy Bridge Xeon E5-2670 2.6GHz cores and 32 GB RAM per node) at ANL, and the Cori supercomputer at NERSC (with 64 Haswell Xeon E5-2698v3 2.3GHz cores and 128 GB RAM per node). Fig. 11 shows that the MBTR workflow consistently outperforms ESMF on both the machines as the number 15    of processes used by the coupler is increased. The timings shown here represent the total remapping time i.e., cumulative computational time for generating the super mesh and the (conservative) remapping weights.

The relatively better scaling for MOAB on the Blues cluster is due to faster hardware and memory bandwidth compared to the Cori machine. The strong scaling efficiency approaches a plateau on Cori Haswell nodes as communication costs for the

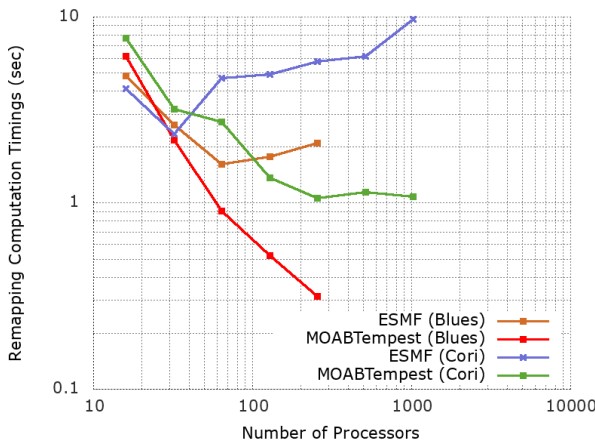

**Figure 11.** CS (E=614400 quads) → CS (E=153600 quads) remapping (-m conserve) on LCRC/ALCF and NERSC machines

coverage mesh computation start dominating the overall remapping processes, especially in the limit of $\frac{nele}{process} \to 1$ at large node counts.

### 4.3.2   Strong Scalability of Spectral Projection (SE→FV)

To further evaluate the characteristics of in-memory remapping computation, along with cost of application of the weights during a transient simulation, a series of further studies were executed on the NERSC Cori system to determine the spectral projection of a real dataset between atmosphere and ocean components in E3SM. The source mesh contains 4th order spectral element temperature data defined on Gauss-Lobatto quadrature nodes (cGLL discretization) of the CS mesh, and the projection is performed on a MPAS polygonal mesh with holes (FV discretization). A direct comparison to ESMF was unfeasible in this study since the traditional workflow requires the computation of a dual mesh transformation of the spectral grid. Hence, only timings for MBTR workflow is shown here.

Two specific cases were considered for this SE→FV strong scaling study with conservation and monotonicity constraints.

1. **Case A (NE30):** 1-degree CS (30 edges per side) SE mesh (nele=5400 quads) with $p = 4$ to MPAS mesh (nele=235160 polygons)

2. **Case B (NE120):** 0.25-degree CS (120 edges per side) SE mesh (nele=86400 quads) with $p = 4$ to MPAS mesh (nele=3693225 polygons)

The performance tests for each of these cases were launched with three different process execution layouts for the atmosphere, ocean components and the coupler.

(a) Fully colocated PE layout: $N_{atm} = N_x$ and $N_{ocn} = N_x$

(b) Disjoint-ATM model PE layout: $N_{atm} = N_x/2$ and $N_{ocn} = N_x$

(c) Disjoint-OCN model PE layout: $N_{atm} = N_x$ and $N_{ocn} = N_x/2$

**Table 1.** Strong scaling on Cori for SE→FV projection with two different resolutions on a fully colocated PE layout

| Number of processors | Case A (NE30) | | Case B (NE120) | |
|---|---|---|---|---|
| | Intersection (sec) | Compute Weights (sec) | Intersection (sec) | Compute Weights (sec) |
| 16 | 0.936846 | 0.64983 | 145.623 | 9.732 |
| 32 | 0.449022 | 0.429028 | 53.1244 | 5.78093 |
| 64 | 0.377767 | 0.373476 | 22.7167 | 4.92151 |
| 128 | 0.255154 | 0.270574 | 6.70485 | 2.79397 |
| 256 | 0.180136 | 0.18272 | 2.26435 | 1.71835 |
| 512 | 0.162388 | 0.104737 | 1.25471 | 0.928622 |
| 1024 | 0.203354 | 0.0932475 | 0.680122 | 0.618943 |

A breakdown of computational time for key tasks on Cori with up to 1024 processes for both the cases is tabulated in Table 1 on a fully colocated decomposition i.e., $N_{ocn} = N_{atm} = N_x$. It is clear that the computation of parallel intersection mesh strong scales well for these production cases, especially for larger mesh resolutions (Case B). For the smaller source and target mesh resolution (Case A), we notice that the intersection time hits a lower bound that is dominated by the computation of the coverage mesh to enclose the target mesh in each task. It is important to stress that this one time setup call to compute remap operator, per component pair, is relatively much cheaper compared to individual component and solver initializations and get amortized over longer transient simulations. It is also worth noting that as the I/O bandwidth in emerging architectures are not scaling in line with the compute throughput, such an online workflow can generally be faster than parallel I/O for reading the weights from file at scale. The MBTR implementation is also flexible to allow loading the weights from file directly in order to preserve the existing coupler process with MCT. In comparison to the computation of the intersection mesh, the time to assemble the remapping weight operator in parallel is generally smaller. Even though both of these operations are performed only once during the setup phase of the E3SM simulation, the weight operator computation involves several validation checks that utilize collective MPI operations, which do destroy the embarrassingly parallel nature of the calculation, once appropriate coverage mesh is determined in each task.

The component-wise breakdown for the advancing front intersection mesh, the parallel communication graph for sending and receiving data between component and coupler, and finally, the remapping weight generation for the SE→FV setup for NE30 and NE120 cases are shown in Fig. 12. The cumulative time for this remapping process is shown to scale linearly for NE120 case, even if the parallel efficiency decreases significantly in the NE30 case, as expected based on the results in Table 1. Note that the MBTR workflow provides a unique capability to consistently and accurately compute SE→FV projection weights

in parallel, without any need for an external pre-processing step to compute the dual mesh (as required by ESMF) or running the entire remapping process in serial (TempestRemap).

Another key aspect of the results shown in Fig. 12 is the relative indifference in performance of the algorithms to the type of PE layout used to partition the component process space. Theoretically, we expect the fully disjoint case to perform the worst, and a full colocated case with maximal overlap to perform the best, since the layout directly affects the total amount of data communicated for both the mesh and field data. However, in practice, with online repartitioning strategies exposed through Zoltan (PHG and RCB), overall scaling of the remapping algorithm is nearly independent of the PE layout for the simulation. This is especially evident from the timings for the coverage mesh computation for the NE120 case for all three PE layouts.

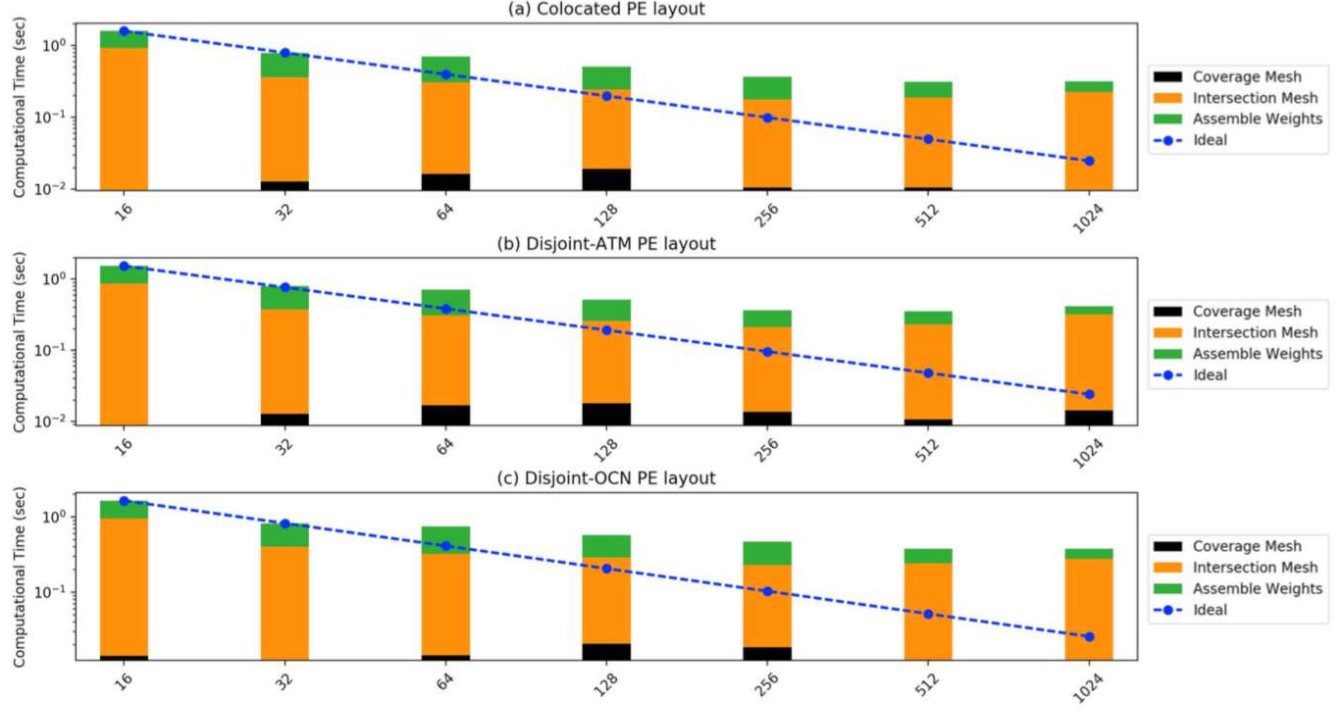

(a) NE30 component-wise strong scaling

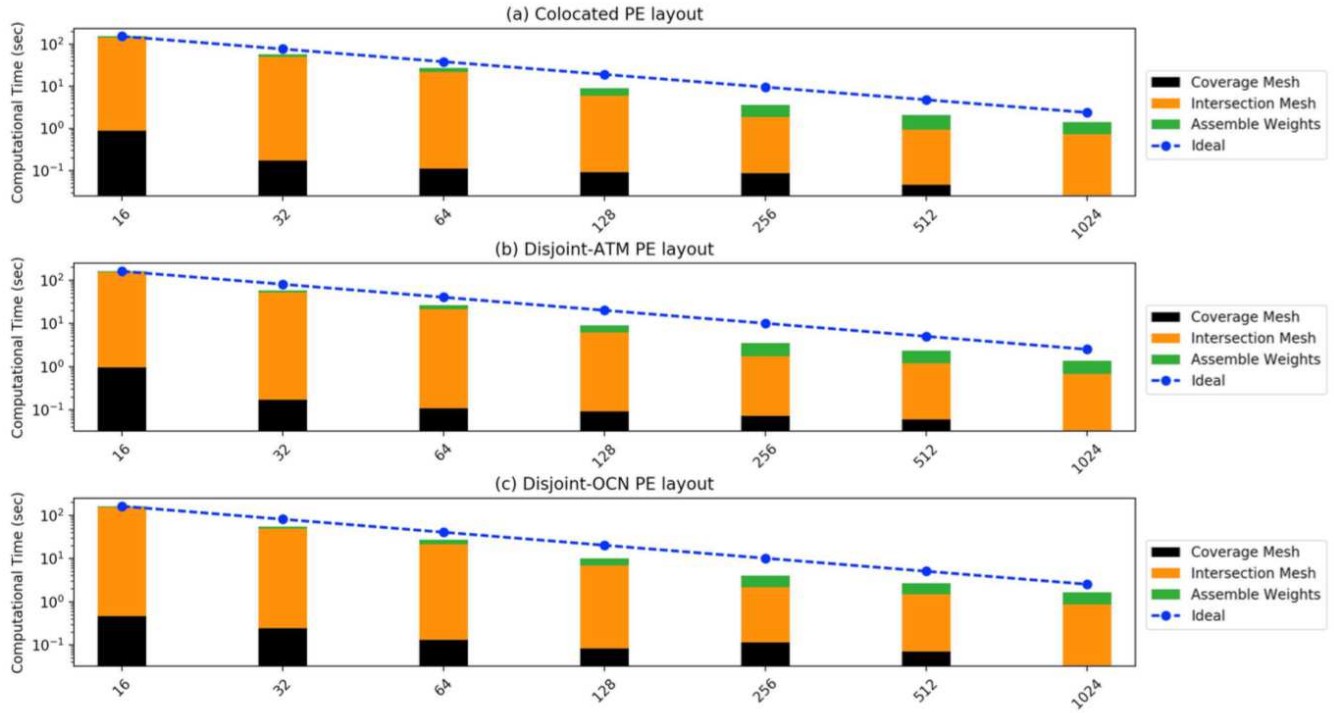

(b) NE120 component-wise strong scaling

**Figure 12.** Strong scaling study for the NE30 and NE120 cases for spectral projection with Zoltan repartitioner on Cori

## 4.4 Effect of partitioning strategy

In order to determine the effect of partitioning strategies described in Fig. 5, the NE120 case with the trivial decomposition and Zoltan geometric partitioner (RCB) were tested in parallel. Fig. 13 compares the two strategies for optimizing the mesh migration from the component to coupler. These strategies play a critical role in task mapping and data locality for the source
coverage mesh computation, in addition to determining the communication graph complexity between the components and the coupler. This comparison highlights that the coverage mesh cost reduces uniformly at scale, while the trivial partitioning scheme behaves better on lower core counts as shown in Fig. 13-(a). The communication of field data between the atmosphere component and the coupler resulting from the partitioning strategy is a critical operation during the transient simulation, and generally stays within network latency limits in Cori (shown in Fig. 13-(b)). Eventhough the communication kernel does not
show ideal scaling on increasing node counts, the relative cost of the operation should be insignificant in comparison to total time spent in individual component solvers. Note that production climate model solvers require multiple data fields to be remapped at every rendezvous timestep, and hence the size of the packed messages may be larger for such simulations. We also note that there is a factor of 3 increase in the communication time to send and receive data, which occurs after the 64 process count on Cori in Fig. 13-(b). This is an artifact of the additional communication latency due to the transition from an
intra-node (each Haswell node in Cori accommodates 64 processes) to inter-node nearest neighbor data transfer when using multiple nodes. In this strong scaling study, the net data size being transferred reduces with increasing core counts, and hence the point-to-point communication beyond 128 processes is primarily dominated by network latency and task placement. As part of future extensions, we will further explore task mapping strategies with the Zoltan2 (Leung et al., 2014) library, in addition to online partition rebalancing to maximize geometric overlap, and minimize communication time during remapping.

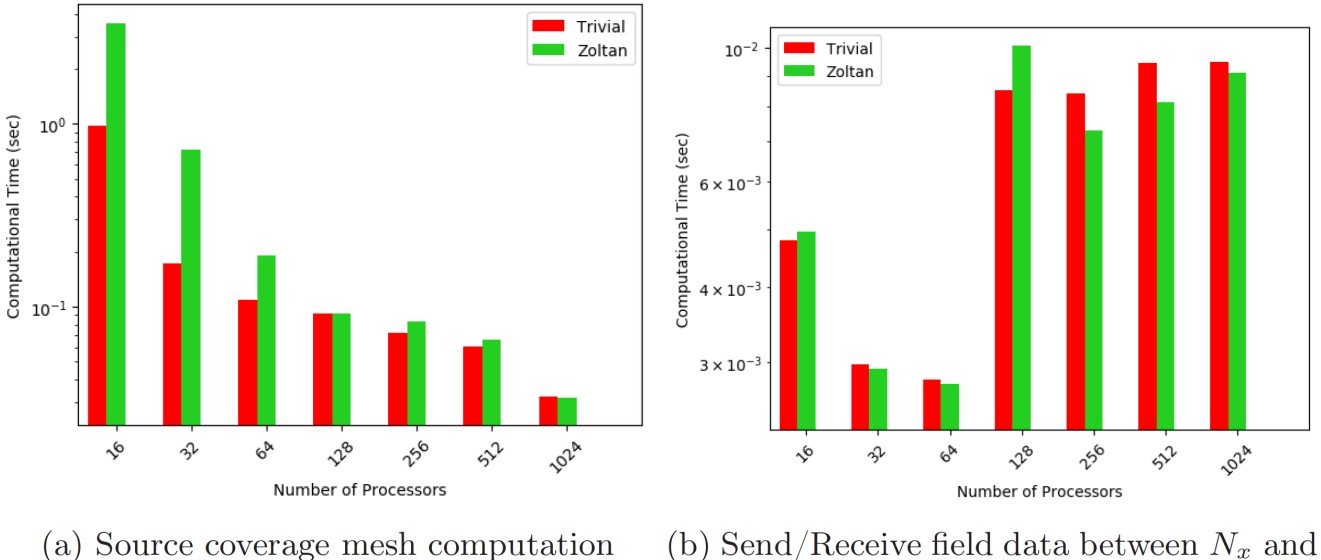

(a) Source coverage mesh computation

(b) Send/Receive field data between $N_x$ and $N_{atm}$

**Figure 13.** Scaling of the communication kernels driven with the parallel graph computed with a trivial redistribution and the Zoltan geometric (RCB) repartitioner for the NE120 case with $N_{ocn} = N_x$ and $N_{atm} = N_x/2$ on Cori

### 4.5 Note on Application of Weights

Generally, operations involving Sparse Matrix-Vector (SpMV) products are memory bandwidth limited (Bell and Garland, 2009), and occur during the application of remapping weights operator on to the source solution field vector, in order to compute the field projection onto the target grid. In addition to the communication of field data shown in Fig. 13-(b), the cost of remapping weight application in parallel (presented in Fig. 14) determines the total cost of the remapping operation during runtime. Except for the case of cGLL target discretizations, the parallel SpMV operation during the weight application do not involve any global collective reductions. In the current E3SM and OASIS3-MCT workflow, these operations are handled by the MCT library. In high resolution simulations of E3SM, the total time for the remapping operation in MCT is primarily dominated by the communication costs based on the communication graph, similar to the MBTR workflow. However, a direct comparison of the communication kernels in these two workflows is not yet possible, since the offline maps for MCT that are generated with ESMF use the "dual" grid, while the online maps generated with MBTR utilize the original spectral grid with no approximations, which results in very different communication graph and non-zero pattern in the remap weight matrices.

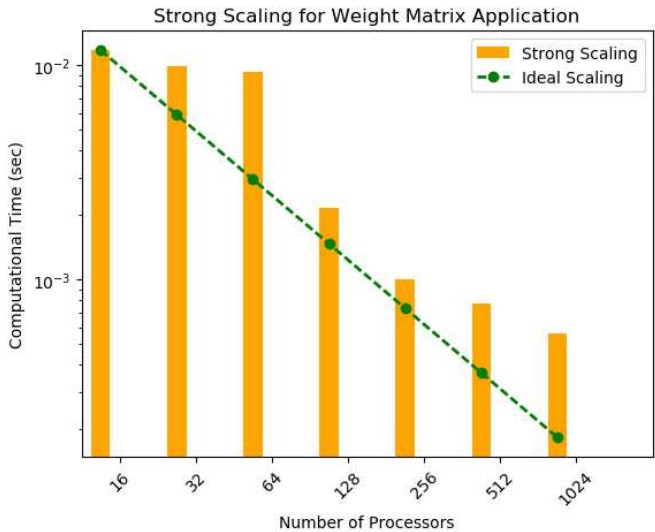

**Figure 14.** SE→FV remapping weight operator application for the NE120 case on Cori

## 5 Conclusion

Understanding and controlling primary sources of errors in a coupled system dynamically, will be key to achieving predictable and verifiable climate simulations on emerging architectures. Traditionally, the computational workflow for coupled climate simulations has involved two distinct steps, with an offline pre-processing phase using remapping tools to generate solution field projection weights (ESMF, TempestRemap, SCRIP), which are then consumed by the coupler to transfer field data between the component grids.

The offline steps include generating grid description files and running the offline tools with the problem-specific options. Additionally many of state-of-science tools such as ESMF and SCRIP require additional steps to specially handle interpolators from SE grids. Such workflows create bottlenecks that do not scale, and can inhibit scientific research productivity. When experimenting with refined grids, a goal for E3SM, this tool chain has to excercised repeatedly. Additionally, when component meshes are dynamically modified, either through mesh adaptivity or dynamical mesh movement to track moving boundaries, the underlying remapping weights must be recomputed on the fly.

To overcome some of these limitations, we have presented scalable algorithms and software interfaces to create a direct component coupling with online regridding and weight generation tools. The remapping algorithms utilize the numerics exposed by TempestRemap, and leverage the parallel mesh handling infrastructure in MOAB to create a scalable in-memory remapping infrastructure that can be integrated with existing coupled climate solvers. Such a methodology invalidates the need for dual grids, preserves higher-order spectral accuracy, and locally conserves the field data, in addition to monotonicity constraints, when transferring solutions between grids with non-matching resolutions.

The serial and parallel performance of the MOAB advancing front intersection algorithm with linear complexity ($O(n)$) was demonstrated for a variety of source and target mesh resolution combinations, and compared with the current state-of-science regridding tools such as ESMF (serial/parallel) and TempestRemap (serial) that have a $O(nlog(n))$ complexity using the Kd-tree datastructure. The MOAB-TempestRemap (MBTR) software infrastructure yields a balance of both the scalable performance on emerging architectures without sacrificing discretization accuracy for component field interpolators. There are also several optimizations in the MBTR algorithms that can be implemented to improve finer-grained parallelism on heterogeneous architectures, and to minimize data movement with better partitioning in combination with load rebalancing strategies. Such a software infrastructure provides a foundation to build a new coupler to replace the current offline-online, hub-and-spoke MCT-based coupler in E3SM, and offer extensions to enable a fully distributed coupling paradigm (without the need for a centralized coupler) to minimize computational bottlenecks in a task-based workflow.

*Code availability.* Information on the availability of source code for the algorithmic infrastructure and models featured in this paper is tabulated below.

| Short name | Code availability |
| --- | --- |
| **E3SM** | E3SM Project (2018) is under active development funded by the US Department of Energy. E3SM version 1.1 has been publicly released under an open-source 3-clause BSD license in August 2018, and available at GitHub. |
| **MOAB** | MOAB Tautges et al. (2004) is an open-source library under the umbrella of the SIGMA toolkit (2014) Mahadevan et al. (2015), and is publicly available under the Lesser GNU Public License (v3) on BitBucket. v5.1.0 was released on Jan 07, 2019 and available here. DOI: 10.5281/zenodo.2584863. |
| **TempestRemap** | The TempestRemap Ullrich and Taylor (2015); Ullrich et al. (2016) source code is available under a BSD open-source license and hosted in GitHub. v2.0.2 was released on Dec 19, 2018 and available here. |

*Video supplement.* The video supplements for the serial and parallel advancing front mesh intersection algorithm to compute the supermesh ($\mathbf{\Omega_S} \bigcup \mathbf{\Omega_T}$) of a source ($\mathbf{\Omega_S}$) and target ($\mathbf{\Omega_T}$) grid is demonstrated.

| Short name | Video description and availability |
|---|---|
| **Serial advancing front mesh intersection** | Intersection between CS and MPAS grids on a single task is illustrated. DOI:10.6084/m9.figshare.7294901 |
| **Parallel advancing front mesh intersection** | Simultaneous parallel Intersection between CS and MPAS grids on two different tasks are illustrated side by side. DOI:10.6084/m9.figshare.7294919 |

*Author contributions.* VM and RJ wrote the paper (with comments from IG and JS). VM and IG designed and implemented the MOAB integration with TempestRemap library, along with exposing the necessary infrastructure for online remapping through iMOAB interfaces. IG and JS configured the MOAB-TempestRemap remapper within E3SM, and verified weight generation to transfer solution fields between atmosphere and ocean component models. VM conducted numerical verification studies and executed both the serial and parallel scalability
studies on Blues and Cori LCF machines to quantify performance characteristics of the remapping algorithms. The broader project idea was conceived by Andy Salinger (SNL), RJ, VM, and IG.

*Competing interests.* The authors declare that they have no conflict of interest.

*Acknowledgements.* Support for this work was provided by the Climate Model Development and Validation - Software Modernization project, and partially by the SciDAC Coupling Approaches for Next Generation Architectures (CANGA) project, which are funded by the
10 US Department of Energy (DOE), Office of Science Biological and Environmental Research Program. CANGA is also funded by the DOE Office of Advanced Scientific Computing Research. This research used resources of the Argonne Leadership Computing Facility at Argonne National Laboratory, which is supported by the Office of Science of the U.S. Department of Energy under contract DE-AC02-06CH11357, and resources of the National Energy Research Scientific Computing Center, a DOE Office of Science User Facility supported by the Office of Science of the U.S. Department of Energy under Contract No. DE-AC02-05CH11231. We gratefully acknowledge the computing
resources provided on Blues, a high-performance computing cluster operated by the Laboratory Computing Resource Center at Argonne National Laboratory. We would also like to thank Dr. Paul Ullrich at University of California, Davis for several helpful discussions regarding remapping schemes and, in particular, implementations in TempestRemap. We would also like to thank the anonymous reviewers for their indepth comments that helped clarify and improve the manuscript.

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
