# Peer review of "Improving climate model coupling through a complete mesh representation: a case study with E3SM (v1) and MOAB (v5.x)"

_Geoscientific Model Development, 2018_

## Short Comment (SC1) · 28 Nov 2018

Dear authors,

in your paper you emphasise the software need for being able to perform an efficient, scalable and parallel online neighbourhood search between source and target grids. In your introduction you discuss the state of the art and mention ESMF and OASIS3-MCT. You argue that these tools are not really suited or are not used in the above mentioned online mode. In this context I am missing missing a discussion of

Liu et al., C-Coupler2: a flexible and user-friendly community coupler for

model coupling and nesting Geosci. Model Dev., 11, 3557-3586, 2018 https://doi.org/10.5194/gmd-11-3557-2018

and

Hanke et al, YAC 1.2.0: new aspects for coupling software in Earth system modelling Geosci. Model Dev., 9, 2755-2769, 2016 https://doi.org/10.5194/gmd-9-2755-2016

Both software products are designed to perform a parallel online search at runtime, and I think that both software tools are already used in this mode in the daily operation of the respective coupled modelling efforts – at least I can confirm that this is the case for YAC within ICON (see e.g. https://mpimet.mpg.de/en/science/projects/integrated-activities)

In my opinion it will be helpful if you can tell us as readers how your effort is related to the above mentioned publications and where your effort is superior to the above, last but not least as they both appeared in the same journal.

Sincerely,

Rene Redler

---

## Referee Comment (RC1) · Anonymous Referee #1 · 9 Dec 2018

General Comments
* * *
The paper describes with accurate details the work accomplished to implement an efficient workflow for remapping and communications tasks in E3SM. The new workflow being based on the MOAB data structures and mesh libraries and on the TempestRemap algorithms, it is identified by the MBTR acronym.

Thourough details on the algorithm and on the implementation are provided and a complete performance analysis is carried on for two test cases.

[Figure]

The work is interesting and deserves publication, yet, in the current formulation, it risks to be a mixture between a technical report and a research paper.

The potential of the proposed workflow should be better situated w.r.t. to other coupled infrastructures than E3SM.

Some suggestions are provided in the "Specific Comments" and "Needed clarifications" sections.

Specific Comments

——————————

The scheme at p.4 ll.7-17 and the following comparison in section 2 should clearly distinguish what features are available in distributed softwares or have just been presented as conceptual algorithms (e.g. the advanced clipping: is it in Portage?) and what has been practically tested by the authors or just inferred from documentations (suggestion: avoid sentences like "It is also unclear whether" unless you add the source of your information. User guide, publications, application cases, ...)

From the user point of view, it is important to know beforehand the amount of information needed to describe the meshes, the decompositions, the fields and the treatments. In the comparison of the coupling approaches this point should be stressed. The MCT paradigm requires a very agile data description (in it's OASIS3 implementation, it is a commitment to be able to work without the connectivity description - at the price of being "oblivious" of some structures). Please assess somehow the user friendliness of the MOAB API's (in particular in their fortran version). A good anchor could be p.8 ll.18-19 where you mention the need of introducing extra calls to describe the details of the mesh to MOAB.

Please include considerations on the memory requirements for storing the MOAB data structures and the supermesh informations. Is there any extra-memory to be accounted for on the source and target processes if adaptive or moving meshes have

to be enrolled runtime in MBTR? This assessment could make the last paragraph of section 3.1 more useful, since its aim is not very clear in the current paper. Refer also to step 5: of Algorithm 1.

Description in section 3 is fluctuating between the hub-and-spoke and the MOAB workflow (e.g. p.10 l.8) please state clearly what's the starting point, the reference for comparison and the new proposal.

The potential of hybrid parallel implementations (MPI processes + threaded tasks) is not always consistently addressed neither in MBTR (e.g. for intersection computation) nor for comparison. Check that the use of process and task is coherent through the whole paper (in particular section 3.4), please.

In order not to restrain the scope of this paper to the replacement of ESMF in E3SM, how would you assess and compare the overall efficiency w.r.t. to a "non hub-and-spoke" coupler interleaving computation and remapping on the same sets of processors (e.g. YAC, OASIS3-MCT) and with couplers already addressing the issue of online weights updates (YAC, C-COUPLER2)?

Needed clarifications

————————

p.4 l.29 explain "consistently respecting the underlying discretization" or remove the sentence.

p.6 l.6 define (or cite) "component architecture". Versus what?

p.6 l.21 does "Fig. 1 (right)" apply to OASIS3-MCT also? Or does the mere-library approach (no separate $N\_x$ for the coupler) defines another workflow?

p.7 l.25 what does "field [...] aware" mean?

p.7 l.29 "during the setup phase" is in contradiction with the aim of allowing for adaptive and moving meshes. Indicate whether it is just a practical choice in the current

implementation.

p.7 ll.30-32 in what exactly is the MBTR stack an improvement w.r.t. the MCT view, since MCT is able to handle decomposed meshes?

p.8 l.3 indicate under which scheduling assumptions the $N_x$ processes can share with the $N_{\{c,l\}}$ processes part of the processor ressources as implied later by Fig. 4.

p.8 l.14 please define a "DoF": since it is not a word used for cell-centered couplers, it is not a common term for all the readers.

p.8 whole section 3.1 (and following) please include references or links for HOMME, MPAS, VisIt and in general do so for all mentioned models, libraries and other software tools (Zoltan, ParMetis, Eigen3, etc).

p.8 l.26 why "replicated" meshes. Isn't it rather "partitioned"?

p.8 l.29 "in terms of a 'Tag'." is a useless statement unless you make it clear to the reader.

p.8 l.29 $n_p$ has not been defined and is not trivial.

p.9 Algorithm 1. The formulation is too compact and missing some previous definition. Insert references to following sections for details.

p.11 ll.2-3 state here (or anticipate) the rational for replacing MCT as a broker.

p.12 l.8 Kd-tree is a relatively common technique (already mentioned at p.4) BVH-tree deserves a reference here (only provided at p.20).

p.12 l.11 why "unique" ?

p.12 l.17 the same consideration as for p.7 l.29 applies.

p.12 ll.26-29 Fig 6. is not immediate to read without some further "step to step" details in the text.

pp.12-13 subsection 3.3.1 does the seed determination can be fully automated or its efficiency depend on user tuning?

p.13 l.11 what does the sentence "without approximations" refer to (especially w.r.t what alternative)?

p.14 l.6 computing a meaningful bounding box is not trivial in polar or periodicity regions for lon/lat grids.

p.14 l.7 does "to all tasks" refer to tasks (or rather processes) on the source side?

p.14 l.8 "Cells [...] are sent": how are they represented? What's the size of the communications? Is any packing strategy used to avoid latency in separate small communications?

p.14 l.10 please clarify the term "superset": a superset usually refers to inclusion of similat objects. Does it imply that the after representation in MOAB - through the definition of the supermesh - the source and the target side share the same spatial discretisation?

p.14 l.14 is the "crystal" router explained in Tautges et al. (2012) [N.B. reference not freely available] or does it need an extra reference?

p.15 l.5 how expensive can be the communication of ghost intersection elements on highly distributed components?

p.15 l.13 does "has the potential to" mean that is just an idea or is there a prototype?

p.16 l.28 "it is non-trivial to": did you find a way?

p.21 l.8 reference to NE11 configuration not known to the reader.

p.26 Fig.14(b) provide an explanation for the difference of behaviour when going beyond 64 processors

Technical Corrections

[Figure]
* * *
p.2 l.5 vs l.31 (and elsewhere) make the use of "donor" or "source" consistent

p.2 l.9 should probably be "conservation for critical quantities"

p.3 l.20 the subject of "that nonlinearly couple" should not be "solution fields" (they are just exchanged in the nonlinear coupling process)

p.6 l.6 unclear (if not useless) reference "Section (1)"

p.6 l.32 the "GLL acronym" is used before definition which is given a few lines later

p.7 l.29 "an in-memory" instead of "a in-memory"

p.8 l.26 remove "a" before "replicated SE and MPAS" meshes

p.9 Algorithm 1.

- Step 1: if l can only be s or t - as in step 4: - indicate l \in [s,t] also in step 1: otherwise if the formulation is generic for more than one mesh for component, the naming should be consistent.

- Step 2: if you indicate W_{ij} instead of W_{st} you should not define i,j as a mesh pair. Later at step 20: i takes a specific meaning.

p.12 l.1 "partitioner" instead of "repartioner"

p.12 l.23 remove "is" before "results"

p.12 l.26 "each" instead of "Each"

p.14 Fig.7 caption: "fully covers" instead of "fully cover"

p.15 l.3 "the intersection vertices [...] need" instead of "needs"

---

## Referee Comment (RC2) · Anonymous Referee #2 · 19 Dec 2018

This paper describes some new algorithms and implementation in E3SM with regard to generation of interpolation weights on diverse and complex grids. It includes a fairly comprehensive set of results and description. It is well organized, well thought-out, and well written. These results seem to leapfrog prior efforts.

General Comments:

The main body text where references are given needs to be reformatted. The references and text are not clearly separated and make them difficult to read. There is also a reference to "Section. (1)" on Page 6, line 6, that I believe suffers from the same formatting problem?

[Figure]

There are clearly trade-offs in generating weights online vs offline. This is highlighted in the paper a number of times with emphasis on the benefits in workflow associated with online capability. High performing online weights generation also has the ability to support non-static grids. Both of these are great benefits of online weights generation. The outstanding questions that are not answered in the paper are (1) can the weights generated online be counted on to produce error free interpolation (conservation, monotonicity, etc) without first being reviewed and validated offline? (2) is the weights generation capability robust and reliable enough to run on different platforms and expect the same results to at least roundoff? (3) is it faster to generate weights online vs reading them in? (4) Is there some benefit to generating the weights online and then being able to reuse them as compared to regenerating them each time the model is run with regard to performance or reproducibility? It would be helpful if the paper addressed these issues if possible. These issues are partly raised in a few places in the paper, at least Page 6, Lines 25-27 and Page 18, Lines 9-10. Some addiitonal discussion/results might be interesting.

Page 8, line 1, what does mesh aware entail? You discuss the potential all-to-all nature of weights generation in the prior paragraph. What information does MOAB carry around that help this problem and how much memory does it require? Does each task have access to the global grid information without requiring communication? Or is there just neighbor connectivity stored? If the grid description is not global on all tasks, how much is communication reduced vs having only local information? The MCT gsmap has global information on each task related to ID, and pe. I assume the MOAB mesh has the same plus coordinate information? The MCT gsmap is generally compressed significantly because the information can be defined via a single start and end ID for certain kinds of decompositions. Since the MOAB mesh carries more info, I assume that compression is not possible and that the mesh consists of "n" fields of data for each gridpoint/corner/edge/etc? Is that a lot of data? Does the memory scale at all at higher resolutions and higher pe counts? I'm sure much of this is documented in MOAB papers, but it would be nice to add a sentence or two about it in this paper.

[Figure]

How the new capabilities are implemented in E3SM is somewhat unclear. In Figure 1, it looks like there is no longer a coupler. Where are the non-coupling non-mapping coupler operations (merging, atm/ocn flux, diagnostics, etc) being computed? In text, it sounds like the coupler component still exists but that the underlying MCT datatypes were swapped for MOAB datatypes, an additional set of calls were added in the component coupling layer to more fully describe the meshes, the online weights generation was added, and the online sparse matrix multiply was converted from MCT calls to MOAB calls. But then at page 17, line 12-15, it sounds like the coupling is between pairs of components excluding a coupler. It would be good if this were clarified.

Specific Comments:

Page 6, line 20 "oas (2018)" ?

Page 7, line 29 fix "a in-line", should be "an in-line"

Page 8, line 7 Alg. 1 -> Algorithm 1

Page 11, Fig 5b. It seems unlikely that the trivial decompostion would be someone's first guess for best performing decomposition with knowledge of how the coupling/mapping work. Having said that, I'm surprised it performs as well as it does in Figure 14. There are lots of other resonable decompositions, why were Trival and Zoltan chosen to be highlighted in this paper? And why does the trivial decomposition perform so well in Figure 14.

Page 12 line 23, remove "is" in "is results in"

Page 16, line 28, bit-for-bit capability is sometimes important to achieve, certainly for identical runs, also for runs on different pe counts (sometimes with a performance penalty via an optional flag). This sentence left me asking what the bit-for-bit capabilities are and what risks are introduced when computing online versus reusing.

Page 19, Fig 9. it would be nice if the scale were not so ad-hoc and instead something more like (0,80,4). Scales like the one shown just make the figure more difficult to
digest and in this case, there is no benefit to have the breaks defined as they are relative to something simpler and easier to read. Also, I'm not sure color adds anything, I think the same could be shown via a contour plot, possibly clearer and simpler still.

Page 18, line 29 "serial runs" vs page 20, line 1 "better performance in MBTempest ... offers avenues to incorporate task level parallelism ...". Are these serial runs or something else? Serial in MPI but using shared memory parallelism? Is that still serial?

Page 20, lines 18-19. For the 1024ˆ3 test case, are weights being generated in 2d or 3d? If 3d, is this test case an order of magnitude (or more) larger than the largest climate model grids? Might be worth clairfying in text.

Page 20, lines 24-26. I agree that the initialization cost is amortized for long production climate runs. But in your example, that init cost is order (hundreds) of seconds (see fig 10c). That is for a single set (pair?) of weights. In coupled climate model, there are often order (10) of these to be done. Now we're talking 1000s of seconds which starts to sound expensive in production but is certainly very expensive for short development test runs. Would it be cheaper to store the weights in a file and read them in the next time? (see general comments). Having said that, please confirm that weights are generated on each of the 1 billion gridcells (1024ˆ3) in 3d. And if so, that's a lot of gridcells.

Page 21, Figure 10, I am struggling to read the axes and other text on the plots

Page 21, Figure 10b shows scaling to 512k pes for a problem size of 1024ˆ3. The final point has 2000 gridcells per process which is still relatively big. What if you chose a problem size of 128ˆ3 or 256ˆ3 and tried to scale to 512k cores?

Page 25, figure 13. Is there benefit to showing the three results (colocated plus two disjoint). The results are very similar for the three cases, at least as presented. And there is no discussion of the differences/similarities in text.

Page 26, figure 14b. I am surprised there is so little scaling of the send/recv at NE120

and the core counts presented. I understand the claim that the absolute cost is small in all cases. I guess you are only redistributing 86k (NE120) elements, maybe that's expected then? The jump between 64 and 128 must be a machine thing, going off-node or something? At 128 cores, you should be transferring over 500 elements per core. Do you expect no scaling beyond that given the message size? Do you want to mention any of this in the paper?

Please confirm that you describe which machine the tests are run on in text and it might be beneficial to include that information in the figure captions. For page 27, figure 15, maybe remind us that it's case B of Table 1 (I think that's correct) in text.

---

## Author Comment (AC1) · 19 Dec 2018

Dear Dr. Redler,

We kindly appreciate the references you provided to similar online remapping efforts that are available as part of both the YAC and the C-Coupler software infrastructures. We have modified the manuscript and updated the background section discussions accordingly. While detailed performance of both YAC and Common Remapping software in C-Coupler2 are not directly available, comparison of the intersection computation approaches and 3-D field interpolation algorithms have been discussed where relevant. We also want to note that TempestRemap exposes a unique capability to compute

field projections between spectral element and other FV/cGLL discretizations without any approximations to the underlying grid (such as using an intermediate dual mesh). The presented work in the manuscript leverages this capability and expands the feature to the parallel setting as well. In contrast, it is our understanding that YAC, CoR and ESMF do not directly handle high-order remapping from SE grids.

We request you to review the updated paper and welcome other comments that would improve the scope of the manuscript.

Best regards,

Vijay Mahadevan

Please also note the supplement to this comment:
https://www.geosci-model-dev-discuss.net/gmd-2018-280/gmd-2018-280-AC1-supplement.pdf

––––––––––––––––––––––––––––––––

**Supplement:**

[revised manuscript text omitted]

---

## Author Comment (AC2) · 24 Jan 2019

Dear Reviewer,

We kindly appreciate the detailed comments and suggestions for modifications to make the manuscript clearer. We have replied specifically to some of the question in the review below, and we are in the process of including the suggested modifications in the final manuscript.

1. Referee: The scheme at p.4 ll.7-17 and the following comparison in section 2 should clearly distinguish what features are available in distributed softwares or

have just been presented as conceptual algorithms (e.g. the advanced clipping: is it in Portage?) and what has been practically tested by the authors or just inferred from documentations (suggestion: avoid sentences like "It is also unclear whether" unless you add the source of your information. User guide, publications, application cases, ...

Author: The advanced clipping features of Portage have been inferred from the publications and we have not tested it for practical climate science remapping applications. We will rephrase the relevant sentences in literature survey.

2. Referee: From the user point of view, it is important to know beforehand the amount of information needed to describe the meshes, the decompositions, the fields and the treatments. In the comparison of the coupling approaches this point should be stressed.

   Author: Yes. We will include the relevant modifications in the text to stress this explicitly.

3. Referee: The MCT paradigm requires a very agile data description (in it's OASIS3 implementation, it is a commitment to be able to work without the connectivity description - at the price of being "oblivious" of some structures). Please assess somehow the user friendliness of the MOAB API's (in particular in their fortran version). A good anchor could be p.8 ll.18-19 where you mention the need of introducing extra calls to describe the details of the mesh to MOAB.

   Author: The MOAB Fortran API exposed through iMOAB interface provides routines to query, create, and manipulate meshes in memory from a native mesh representation in the component models. MOAB can work with a full mesh description and also with a notion of point clouds when needed. As you mentioned, extra calls would be needed to expose the mesh in MOAB format and we will include appropriate modifications to make this clear.

4. Referee: Please include considerations on the memory requirements for storing the MOAB data structures and the supermesh informations. Is there any extra-memory to be accounted for on the source and target processes if adaptive or moving meshes have to be enrolled runtime in MBTR? This assessment could make the last paragraph of section 3.1 more useful, since its aim is not very clear in the current paper. Refer also to step 5: of Algorithm 1.

Author: The adaptive refinement provides localized changes in the mesh database and hence intersection mesh computation along with remapping weights are typically contained with a local compact support region during re-computation. The adaptive mesh modifications are imposed with new vertices and connectivity information. Additionally, apart from the additional memory required for the new DoF numbering and fields, MOAB does not have to explicitly store parent-child additional information in the remapping workflow unless more advanced constrained conservation techniques are required by the components.

5. Referee: Description in section 3 is fluctuating between the hub-and-spoke and the MOAB workflow (e.g. p.10 l.8) please state clearly what's the starting point, the reference for comparison and the new proposal.

Author: We have received several comments about the description in this section. We will make appropriate modifications to make this clearer.

6. Referee: The potential of hybrid parallel implementations (MPI processes + threaded tasks) is not always consistently addressed neither in MBTR (e.g. for intersection computation) nor for comparison. Check that the use of process and task is coherent through the whole paper (in particular section 3.4), please

Author: We will clarify the hybrid parallelism references throughout the manuscript.

7. Referee: In order not to restrain the scope of this paper to the replacement of ESMF in E3SM, how would you assess and compare the overall efficiency w.r.t.

to a "non hub-andspoke" coupler interleaving computation and remapping on the same sets of processors (e.g. YAC, OASIS3-MCT) and with couplers already addressing the issue of online weights updates (YAC, C-COUPLER2)?

Author: This is a harder comparison to make in terms of overall efficiency and performance characteristics without running the couplers on the same set of input grids to generate conservative remapping weights. We are open to suggestions in this front if there are ways we can add value to the manuscript with relevant comparisons.

We welcome any additional comments on this topic although we realize that the discussions are closed at this point.

Sincerely,

Vijay Mahadevan, Robert Jacob, Iulian Grindeanu, Jason Sarich

---

## Author Comment (AC3) · 24 Jan 2019

Dear Reviewer,

We sincerely appreciate the detailed comments and suggestions for modifications to make the manuscript clearer. We have replied specifically to some of the question in the review below, and we are in the process of including the suggested modifications in the final manuscript.

1. Reviewer: The outstanding questions that are not answered in the paper are

(a) can the weights generated online be counted on to produce error free inter-
polation (conservation, monotonicity, etc) without first being reviewed and
validated offline?

(b) is the weights generation capability robust and reliable enough to run on
different platforms and expect the same results to at least roundoff?

(c) is it faster to generate weights online vs reading them in?

(d) Is there some benefit to generating the weights online and then being able
to reuse them as compared to regenerating them each time the model is run
with regard to performance or reproducibility?

It would be helpful if the paper addressed these issues if possible. These issues
are partly raised in a few places in the paper, at least Page 6, Lines 25-27 and
Page 18, Lines 9-10. Some addiitonal discussion/results might be interesting.

Author: Thank you for the detailed comments. Some comments below to address
concerns. We will also add specific information on having a reproducible workflow
using the online remapping implementation.

(a) The online remapping weights use the exact same input grids, discretiza-
tion specifications and even most of the same routines as the offline method
with TempestRemap. These are also exposed directly through a MOAB-
TempestRemap tool (mbtempest) that can be run offline, and in parallel, for
verification and validation along with ability to write out the weights to file.
MOAB handles the mesh decomposition, the parallel intersection computa-
tion, DoF management, and offloads the actual remapping weight compu-
tation to TempestRemap, which has been verified and validated independ-
ently.

(b) Yes the online weight generation workflow is robust and the MOABTR work-
flow works consistently in serial (OSX, Linux) and in parallel on clusters and
large-scale machines (Blues, Cori, Theta).

(c) We have not yet explicitly performed production-case comparison tests against serial I/O read of the weights file along with broadcast to tasks in pes from a single process vs the fully online remapping weight computation with MBTR.

(d) At large scale, we expect the online remapper to be much faster, since I/O is expected to be the slowest component in next generation architectures. Hence, it may be advantageous to run the onine remapper all the time for production runs, after sufficient verification and validation. We can write out the generated weights for provenance and reproducibility.

2. Reviewer: Page 8, line 1, what does mesh aware entail? You discuss the potential all-to-all nature of weights generation in the prior paragraph. What information does MOAB carry around that help this problem and how much memory does it require? Does each task have access to the global grid information without requiring communication? Or is there just neighbor connectivity stored?

Author: Mesh aware indicates that the data-structure has the knowledge to traverse the underlying discrete grid used in the component model. The all-to-all communication may happen if the source and target grids on the coupler PEs have no "geometrically coincident" elements. This means that in order to compute the intersection mesh on a particular task, we need to bring the grid elements and vertices that fully cover the target regions. We explain this task as the source coverage mesh computation in Algorithm 1.

MOAB does not explicitly store datastructures that have a O(P) dependency, where P is the number of processes. Only neighbor connectivity information and "ownership" information for shared entities across process domains is stored.

3. Reviewer: If the grid description is not global on all tasks, how much is communication reduced vs having only local information? The MCT gsmap has global information on each task related to ID, and pe. I assume the MOAB mesh has

the same plus coordinate information? The MCT gsmap is generally compressed significantly because the information can be defined via a single start and end ID for certain kinds of decompositions. Since the MOAB mesh carries more info, I assume that compression is not possible and that the mesh consists of "n" fields of data for each gridpoint/corner/edge/etc? Is that a lot of data? Does the memory scale at all at higher resolutions and higher pe counts?

Author: MOAB is a fully distributed datastructure. MOAB does not have any global storage resembling or similar to the MCT gsmap. We use a transient datastructure with bounding boxes (that encompasses the local elements/vertices) to determine the communication pattern between arbitrary decompositions during the setup phase. All communications from thereon are point-to-point. The key differentiating factor, as you pointed out is that MCT has no notion of the global mesh/vertex distributions and hence optimizations based on topology are unavailable. However, since all meshes are treated as truly unstructured, MOAB does store the 'local' vertex coordinates and the connectivity information without any compression, although in a contiguous array-based datastruture.

The fields are stored contiguously per element and the user manages the DoF layout definition on each local element. So for a SE discretization, DoFs may have canonical numbering with p=4, which results in 16 DoFs. For FV, there may only be 1 data per element. This memory requirement is directly proportional to the solution data vector in each component. MOAB being a general mesh library, does allow users to define data on edges/faces etc but we do not use this functionality in our current coupler implementation. At larger resolutions, each task gets a smaller piece of the mesh along with a smaller footprint of the data as well since MOAB is a fully parallel mesh.

4. Reviewer: How the new capabilities are implemented in E3SM is somewhat unclear. In Figure 1, it looks like there is no longer a coupler. Where are the non-coupling non-mapping coupler operations (merging, atm/ocn flux, diagnostics, etc) being computed? In text, it sounds like the coupler component still exists but that the underlying MCT datatypes were swapped for MOAB datatypes, an additional set of calls were added in the component coupling layer to more fully describe the meshes, the online weights generation was added, and the online sparse matrix multiply was converted from MCT calls to MOAB calls. But then at page 17, line 12-15, it sounds like the coupling is between pairs of components excluding a coupler. It would be good if this were clarified.

Author: The hub coupler still exists. We are currently duplicating the MCT calls alongside the MOAB based coupler in order to fully verify and validate both the accuracy and performance at runtime. After full validation, the MCT coupler will be completely removed from E3SM. The MOAB coupler allows the possibility for ATM to directly compute the remapping weights to project field data to OCN since the intersection will then be carried out through migration of OCN mesh to ATM pes. Hence this pair-wise coupling leads to a more distributed coupling strategy in the future. However, we do envision that there will still be a thin layer of a global coupler, even in the distributed case, to drive the subcycling, to compute merging with weighted combinations of fluxes, for validation and other diagnostics data outputs. We understand that Fig. (1) is somewhat misleading in this context and intend to make modifications to make it clearer.

We welcome any additional comments on this topic although we realize that the discussions are closed at this point.

Sincerely,

Vijay Mahadevan, Robert Jacob, Iulian Grindeanu, Jason Sarich
* * *

---

## Author Comment (AC5) · 5 Mar 2019

Dear Reviewer,

Once again, we thank you for the extensive review comment on the manuscript. Please find our detailed responses for all the issues raised in your review comments. We have included most of the suggested changes in the manuscript, and will be uploading the latest copy to the website. Rebuttals for specific questions have also been included in the response here and we hope that it will provide better context in light of the new changes added.

[Figure]

**GMDD**

Interactive
comment

**1 Needed clarifications**

- Referee: p.4 l.29 explain "consistently respecting the underlying discretization" or remove the sentence.
  Authors: Modified. Please review.

- Referee: p.6 l.6 define (or cite) "component architecture". Versus what?
  Authors: The following reference has been added.
  Zhou, S. J. "Coupling climate models with the earth system modeling framework and the common component architecture." Concurrency and Computation: Practice and Experience 18.2 (2006): 203-213.

- Referee: p.6 l.21 does "Fig. 1 (right)" apply to OASIS3-MCT also? Or does the mere-library approach (no separate $N_x$ for the coupler) defines another workflow?
  Authors: Yes. We could have $N_x = N_{atm}$ and still have Fig. (1) (left), the hub-and-spoke model, just to be clear. But what the distributed coupled model refers to is that the components can directly communicate with each other without an additional hop through the coupler (hence no explicit $N_x$). This is much more efficient in terms of total reduced data-transfers and optimizations that can be performed on pair-wise meshes, which are not available on a many-to-many scenario through a global coupler. In the workflow we have defined in the manuscript, the MBTR workflow allows both.

- Referee: p.7 l.25 what does "field [...] aware" mean?
  Authors: Being field aware indicates that regridder needs to understand the discretization types. Our aim is to provide an online remapper that supports consistent and conservative projection of **FV, cGLL, dGLL** source field data to target meshes, and fill the gap with ESMF based offline remapping workflows that require dual meshes and only support FV-type discretizations.

- Referee: p.7 l.29 "during the setup phase" is in contradiction with the aim of allowing for adaptive and moving meshes. Indicate whether it is just a practical choice in the current implementation.
  Authors: In the current implementation, we are computing the remapping operators only in the setup phase since most existing E3SM workflows do not support adaptive grids. But the MBTR workflow can fully support remapping with moving meshes by recomputing the weight matrices at run-time. The text has been slightly rephrased to clarify this.

- Referee: p.7 ll.30-32 in what exactly is the MBTR stack an improvement w.r.t. the MCT view, since MCT is able to handle decomposed meshes?
  Authors: MBTR is an improvement because it also stores the connectivity of the mesh. In particular, MOAB knows that the neighbor of a point might be on another processor. MCT does not have any of that information, which is essential when performing online remapping computation or performance optimizations/load rebalancing based on mesh topology.

- Referee: p.8 l.3 indicate under which scheduling assumptions the $N_x$ processes can share with the $N_{c,l}$ processes part of the processor resources as implied later by Fig. 4.
  Authors: Sharing resources would be appropriate when the physics of the system requires that a calculation performed in the coupler must happen before the solver in the next component is invoked. Figure 4 doesn't indicate that the coupler is sometimes invoked multiple times as individual components are executed, and it specifically doesn't show the global "driver" layer which controls the overall flow of execution and data transfer.

- Referee: p.8 l.14 please define a "DoF": since it is not a word used for cell-centered couplers, it is not a common term for all the readers.
  Authors: DoF has been expanded.

[Figure]

- Referee: p.8 whole section 3.1 (and following) please include references or links for HOMME, MPAS, VisIt and in general do so for all mentioned models, libraries and other software tools (Zoltan, ParMetis, Eigen3, etc).
  Authors: These references have now been added.

- Referee: p.8 l.26 why "replicated" meshes. Isn't it rather "partitioned"?
  Authors: Corrected.

- Referee: p.8 l.29 "in terms of a 'Tag'." is a useless statement unless you make it clear to the reader.
  Authors: Small description of a tag has been added.

- Referee: p.8 l.29 $n_p$ has not been defined and is not trivial.
  Authors: This sentence has been rephrased

- Referee: p.9 Algorithm 1. The formulation is too compact and missing some previous definition. Insert references to following sections for details.
  Authors: Appropriate references to other sections have now been added.

- Referee: p.11 ll.2-3 state here (or anticipate) the rational for replacing MCT as a broker.
  Authors: We expect MOAB to perform data transfers faster an more efficiently (fewer overall messages) then MCT at scale because of MOAB's crystal router. MOAB will also have better memory scaling because, unlike MCT, it does not have datatypes that can grow with grid or processor size. Finally MOAB will allow a simplified workflow by removing the need for directories of mapping weight files.

- Referee: p.12 l.8 Kd-tree is a relatively common technique (already mentioned at p.4) BVH-tree deserves a reference here (only provided at p.20).
  Authors: Added references for both tree structures

- Referee: p.12 l.11 why "unique" ?
  Authors: Removed. It is clearer now.

- Referee: p.12 l.17 the same consideration as for p.7 l.29 applies.
  Authors: Removed reference to the setup phase.

- Referee: p.12 ll.26-29 Fig 6. is not immediate to read without some further "step to step" details in the text.
  Authors: This comment is unclear. Should the advancing front algorithm be explained better ? We have added references to the front intersection video illustrations that are added as supplementary materials.

- Referee: pp.12-13 subsection 3.3.1 does the seed determination can be fully automated or its efficiency depend on user tuning?
  Authors: The determination is fully automated. However, there may be cases with failures when dealing with meshes with holes where a seed in say an atmosphere mesh may not be able to find a corresponding element containing point in the MPAS mesh (if it falls in a land geographical area). Such cases could require more than one attempt in each partition to get the front computation started.

- Referee: p.13 l.11 what does the sentence "without approximations" refer to (especially w.r.t what alternative)?
  Authors: The sentence "without approximation" refers to the fact that the intersection can be computed to machine precision as the edges become straight lines in a gnomonic plane (projected from great circle arcs on a sphere). If curves on a sphere are not great circle arcs (splines, for example), the intersection between those curves has to be computed using some nonlinear iterations such as Newton Raphson for example (depending on the representation of the curve).

We wanted to indicate in the mauscript that intersection in gnomonic plane is simple to do and "exact"; When you have more general curves on a sphere, you

might even have multiple points of intersections, which could test the robustness and stability of the intersection algorithm. However, note that a latitude arc can intersect a great circle arc in 2 places (this can still be computed exactly to machine precision, without any approximations coming from an iteration).

- Referee: p.14 l.6 computing a meaningful bounding box is not trivial in polar or periodicity regions for lon/lat grids.
  Authors: MOAB stores explicit 3-d bounding boxes, since it is a general mesh query/manipulation library. We have not particularly encountered difficulties in handling lat/lon grids.

- Referee: p.14 l.7 does "to all tasks" refer to tasks (or rather processes) on the source side?
  Authors: This refers to processes on the coupler processing elements. The source/target meshes are already in coupler PEs, and a coverage mesh is computed by appropriately moving only elements required to completely cover target elements in current process.

- Referee: p.14 l.8 "Cells [...] are sent": how are they represented? What's the size of the communications? Is any packing strategy used to avoid latency in separate small communications?
  Authors: MOAB utilizes the aggregated crystal router to efficiently send small data between processes. In an all-to-all communication strategy, with log(N) steps of communication, all the processes get access to the data they need. This is used once during the setup phase to establish point-to-point communication links, which is then used later to pack and send data directly.

During the field transfer from components to coupler, we pack multiple fields together in a single array to send the data to coupler, apply weight matrices on the vectors and transmit back the fields (in a packed and aggregated fashion) to the

target component. The size of such communication is on the order of DoFs in source + target.

- Referee: p.14 l.10 please clarify the term "superset": a superset usually refers to inclusion of similat objects. Does it imply that the after representation in MOAB - through the definition of the supermesh - the source and the target side share the same spatial discretisation?
  Authors: The MOAB view of the supermesh includes the union of all vertices and (elements formed by) edges in both source and target grids. Hence the supermesh is typically the superset of either the source or the target grid. This is only with respect to the actual topology of the grid, and has no correlation to the underlying discretization of field data.

- Referee: p.14 l.14 is the "crystal" router explained in Tautges et al. (2012) [N.B. reference not freely available] or does it need an extra reference?
  Authors: Added.

- Referee: p.15 l.5 how expensive can be the communication of ghost intersection elements on highly distributed components?
  Authors: The communication is typically among nearest neighbors and requires 1-2 rings of elements on average depending on the relative resolution between source and target grids. Since these are direct nearest neighbor computations that are performed only once during the setup phase, the actual impact on overall runtime is small.

- Referee: p.15 l.13 does "has the potential to" mean that is just an idea or is there a prototype?
  Authors: This is an idea and a work in progress at the moment. We expect to spend more time hardening the implementation in the coming year.

- Referee: p.16 l.28 "it is non-trivial to": did you find a way?

Authors: There are ways that could ensure bit-for-bit reproducibility at the cost of heavy sub-optimizations. There are internal discussions to better understand whether the non-BFB algorithmic parts can be isolated together.

- Referee: p.21 l.8 reference to NE11 configuration not known to the reader.
  Authors: NE refers to the number of elements on an edge of a cubed-sphere grid. This has been clarified in the manuscript.

- Referee: p.26 Fig.14(b) provide an explanation for the difference of behaviour when going beyond 64 processors
  Authors: This was an interesting transition in the communication timings as we expanded from intra-node to inter-node regime on Cori that has 64 Haswell cores per node. The overall message passing latency as we cross the 64-core barrier is certainly evident in the figure, especially since we are only communicating one solution data field from the component to coupler and vice-versa. We have added some text in the revised paper to discuss this further.

**2 Technical Corrections**

- Referee: p.2 l.5 vs l.31 (and elsewhere) make the use of "donor" or "source" consistent p.2 l.9 should probably be "conservation for critical quantities" p.3 l.20 the subject of "that nonlinearly couple" should not be "solution fields" (they are just exchanged in the nonlinear coupling process)
  Authors: Done

- Referee: p.6 l.6 unclear (if not useless) reference "Section (1)"
  Authors: Introduced a subsection 1.1 to clarify.

- Referee: p.6 l.32 the "GLL acronym" is used before definition which is given a

few lines later
Authors: Done

- Referee: p.7 l.29 "an in-memory" instead of "a in-memory" p.8 l.26 remove "a" before "replicated SE and MPAS" meshes
  Authors: Removed "a"

- Referee: p.9 Algorithm 1. - Step 1: if l can only be s or t - as in step 4: - indicate $l \in [s,t]$ also in step 1: otherwise if the formulation is generic for more than one mesh for component, the naming should be consistent. - Step 2: if you indicate $W_{ij}$ instead of $W_{st}$ you should not define i,j as a mesh pair. Later at step 20: i takes a specific meaning.
  Authors: Revised and included suggested changes

- Referee: p.12 l.1 "partitioner" instead of "repartioner" p.12 l.23 remove "is" before "results" p.12 l.26 "each" instead of "Each" p.14 Fig.7 caption: "fully covers" instead of "fully cover" p.15 l.3 "the intersection vertices [...] need" instead of "needs"
  Authors: Done

We request you to review the updated paper when it becomes available, and we welcome any further comments that would improve the scope of the manuscript. Best regards,

Vijay Mahadevan
* * *

---

## Author Response (AR1)

**Author's consolidated review comment response on* "Improving climate model coupling through a complete mesh representation: a case study with E3SM (v1) and MOAB (v5.x)" *by* Vijay S. Mahadevan et al.**

mahadevan@anl.gov

**1 Rebuttals for Reviewer 1 comments**

**1.1 Needed clarifications**

- Referee: p.4 l.29 explain "consistently respecting the underlying discretization" or remove the sentence.
  Authors: Modified. Please review.
  Manuscript Diff: Changes in p5. l.9.

- Referee: p.6 l.6 define (or cite) "component architecture". Versus what?
  Authors: The following reference has been added.
  Zhou, S. J. "Coupling climate models with the earth system modeling framework and the common component architecture." Concurrency and Computation: Practice and Experience 18.2 (2006): 203-213.
  Manuscript Diff: Changes in p5. l.28.

- Referee: p.6 l.21 does "Fig. 1 (right)" apply to OASIS3-MCT also? Or does the mere-library approach (no separate $N_x$ for the coupler) defines another workflow?

Authors: Yes. We could have $N_x = N_{atm}$ and still have Fig. (1) (left), the hub-and-spoke model, just to be clear. But what the distributed coupled model refers to is that the components can directly communicate with each other without an additional hop through the coupler (hence no explicit $N_x$). This is much more efficient in terms of total reduced data-transfers and optimizations that can be performed on pair-wise meshes, which are not available on a many-to-many scenario through a global coupler. In the workflow we have defined in the manuscript, the MBTR workflow allows both.

Manuscript Diff: Changes in p6. l.21.

- Referee: p.7 l.25 what does "field [...] aware" mean?
  Authors: Being field aware indicates that regridder needs to understand the discretization types. Our aim is to provide an online remapper that supports consistent and conservative projection of **FV, cGLL, dGLL** source field data to target meshes, and fill the gap with ESMF based offline remapping workflows that require dual meshes and only support FV-type discretizations.
  Manuscript Diff: No changes.

- Referee: p.7 l.29 "during the setup phase" is in contradiction with the aim of allowing for adaptive and moving meshes. Indicate whether it is just a practical choice in the current implementation.
  Authors: In the current implementation, we are computing the remapping operators only in the setup phase since most existing E3SM workflows do not support adaptive grids. But the MBTR workflow can fully support remapping with moving meshes by recomputing the weight matrices at run-time. The text has been slightly rephrased to clarify this.
  Manuscript Diff: Changes in p8. l.30-l.31.

- Referee: p.7 ll.30-32 in what exactly is the MBTR stack an improvement w.r.t. the

MCT view, since MCT is able to handle decomposed meshes?
Authors: MBTR is an improvement because it also stores the connectivity of the mesh. In particular, MOAB knows that the neighbor of a point might be on another processor. MCT does not have any of that information, which is essential when performing online remapping computation or performance optimizations/load rebalancing based on mesh topology.
Manuscript Diff: No changes.

- Referee: p.8 l.3 indicate under which scheduling assumptions the $N_x$ processes can share with the $N_{c,l}$ processes part of the processor resources as implied later by Fig. 4.
Authors: Sharing resources would be appropriate when the physics of the system requires that a calculation performed in the coupler must happen before the solver in the next component is invoked. Figure 4 doesn't indicate that the coupler is sometimes invoked multiple times as individual components are executed, and it specifically doesn't show the global "driver" layer which controls the overall flow of execution and data transfer.
Manuscript Diff: No changes.

- Referee: p.8 l.14 please define a "DoF": since it is not a word used for cell-centered couplers, it is not a common term for all the readers.
Authors: DoF has been expanded.
Manuscript Diff: Changes in p9. l.26.

- Referee: p.8 whole section 3.1 (and following) please include references or links for HOMME, MPAS, VisIt and in general do so for all mentioned models, libraries and other software tools (Zoltan, ParMetis, Eigen3, etc).
Authors: These references have now been added.
Manuscript Diff: Changes in p9. l.29-l.30, p11. l.9.

- Referee: p.8 l.26 why "replicated" meshes. Isn't it rather "partitioned"?

Authors: Corrected.
Manuscript Diff: Changes in p11. l.8-l.9.

- Referee: p.8 l.29 "in terms of a 'Tag'." is a useless statement unless you make it clear to the reader.
  Authors: Small description of a tag has been added.
  Manuscript Diff: Changes in p11. l.11-l.13.

- Referee: p.8 l.29 $n_p$ has not been defined and is not trivial.
  Authors: This sentence has been rephrased
  Manuscript Diff: Changes in p11. l.13-l.15.

- Referee: p.9 Algorithm 1. The formulation is too compact and missing some previous definition. Insert references to following sections for details.
  Authors: Appropriate references to other sections have now been added.
  Manuscript Diff: Added several references to sections and relevant citations in p10. Additional descriptions as needed.

- Referee: p.11 ll.2-3 state here (or anticipate) the rational for replacing MCT as a broker.
  Authors: We expect MOAB to perform data transfers faster an more efficiently (fewer overall messages) then MCT at scale because of MOAB's crystal router. MOAB will also have better memory scaling because, unlike MCT, it does not have datatypes that can grow with grid or processor size. Finally MOAB will allow a simplified workflow by removing the need for directories of mapping weight files.
  Manuscript Diff: No changes.

- Referee: p.12 l.8 Kd-tree is a relatively common technique (already mentioned at p.4) BVH-tree deserves a reference here (only provided at p.20).
  Authors: Added references for both tree structures
  Manuscript Diff: p.14 l.8-9.

- Referee: p.12 l.11 why "unique" ?
  Authors: Removed. It is clearer now.
  Manuscript Diff: p.14 l.12.

- Referee: p.12 l.17 the same consideration as for p.7 l.29 applies.
  Authors: Removed reference to the setup phase.
  Manuscript Diff: p.14 l.18-l.19.

- Referee: p.12 ll.26-29 Fig 6. is not immediate to read without some further "step to step" details in the text.
  Authors: This comment is unclear. Should the advancing front algorithm be explained better ? We have added references to the front intersection video illustrations that are added as supplementary materials.
  Manuscript Diff: No changes.

- Referee: pp.12-13 subsection 3.3.1 does the seed determination can be fully automated or its efficiency depend on user tuning?
  Authors: The determination is fully automated. However, there may be cases with failures when dealing with meshes with holes where a seed in say an atmosphere mesh may not be able to find a corresponding element containing point in the MPAS mesh (if it falls in a land geographical area). Such cases could require more than one attempt in each partition to get the front computation started.
  Manuscript Diff: No changes.

- Referee: p.13 l.11 what does the sentence "without approximations" refer to (especially w.r.t what alternative)?
  Authors: The sentence "without approximation" refers to the fact that the intersection can be computed to machine precision as the edges become straight lines in a gnomonic plane (projected from great circle arcs on a sphere). If curves on a sphere are not great circle arcs (splines, for example), the intersection be-

tween those curves has to be computed using some nonlinear iterations such as Newton Raphson for example (depending on the representation of the curve).

We wanted to indicate in the mauscript that intersection in gnomonic plane is simple to do and "exact"; When you have more general curves on a sphere, you might even have multiple points of intersections, which could test the robustness and stability of the intersection algorithm. However, note that a latitude arc can intersect a great circle arc in 2 places (this can still be computed exactly to machine precision, without any approximations coming from an iteration).

Manuscript Diff: No changes.

- Referee: p.14 l.6 computing a meaningful bounding box is not trivial in polar or periodicity regions for lon/lat grids.
  Authors: MOAB stores explicit 3-d bounding boxes, since it is a general mesh query/manipulation library. We have not particularly encountered difficulties in handling lat/lon grids.
  Manuscript Diff: No changes.

- Referee: p.14 l.7 does "to all tasks" refer to tasks (or rather processes) on the source side?
  Authors: This refers to processes on the coupler processing elements. The source/target meshes are already in coupler PEs, and a coverage mesh is computed by appropriately moving only elements required to completely cover target elements in current process.
  Manuscript Diff: No changes added. Description in rebuttal is detailed.

- Referee: p.14 l.8 "Cells [...] are sent": how are they represented? What's the size of the communications? Is any packing strategy used to avoid latency in separate small communications?
  Authors: MOAB utilizes the aggregated crystal router to efficiently send small data between processes. In an all-to-all communication strategy, with log(N)

steps of communication, all the processes get access to the data they need. This is used once during the setup phase to establish point-to-point communication links, which is then used later to pack and send data directly.

During the field transfer from components to coupler, we pack multiple fields together in a single array to send the data to coupler, apply weight matrices on the vectors and transmit back the fields (in a packed and aggregated fashion) to the target component. The size of such communication is on the order of DoFs in source + target.
Manuscript Diff: No changes added. Description in rebuttal is detailed.

- Referee: p.14 l.10 please clarify the term "superset": a superset usually refers to inclusion of similat objects. Does it imply that the after representation in MOAB - through the definition of the supermesh - the source and the target side share the same spatial discretisation?
  Authors: The MOAB view of the supermesh includes the union of all vertices and (elements formed by) edges in both source and target grids. Hence the supermesh is typically the superset of either the source or the target grid. This is only with respect to the actual topology of the grid, and has no correlation to the underlying discretization of field data.
  Manuscript Diff: No changes added. Description in rebuttal is detailed.

- Referee: p.14 l.14 is the "crystal" router explained in Tautges et al. (2012) [N.B. reference not freely available] or does it need an extra reference?
  Authors: Added.
  Manuscript Diff: p.16 l.21.

- Referee: p.15 l.5 how expensive can be the communication of ghost intersection elements on highly distributed components?
  Authors: The communication is typically among nearest neighbors and requires 1-2 rings of elements on average depending on the relative resolution between

source and target grids. Since these are direct nearest neighbor computations that are performed only once during the setup phase, the actual impact on overall runtime is small.
Manuscript Diff: p.16 l.21.

- Referee: p.15 l.13 does "has the potential to" mean that is just an idea or is there a prototype?
  Authors: This is an idea and a work in progress at the moment. We expect to spend more time hardening the implementation in the coming year.
  Manuscript Diff: No changes added.

- Referee: p.16 l.28 "it is non-trivial to": did you find a way?
  Authors: There are ways that could ensure bit-for-bit reproducibility at the cost of heavy sub-optimizations. There are internal discussions to better understand whether the non-BFB algorithmic parts can be isolated together.
  Manuscript Diff: No changes added.

- Referee: p.21 l.8 reference to NE11 configuration not known to the reader.
  Authors: NE refers to the number of elements on an edge of a cubed-sphere grid. This has been clarified in the manuscript.
  Manuscript Diff: p.23 l.1, p.25 l.4, p.25 l.6

- Referee: p.26 Fig.14(b) provide an explanation for the difference of behaviour when going beyond 64 processors
  Authors: This was an interesting transition in the communication timings as we expanded from intra-node to inter-node regime on Cori that has 64 Haswell cores per node. The overall message passing latency as we cross the 64-core barrier is certainly evident in the figure, especially since we are only communicating one solution data field from the component to coupler and vice-versa. We have added some text in the revised paper to discuss this further.
  Manuscript Diff: p.28 l.13-l.16

**1.2 Technical Corrections**

- Referee: p.2 l.5 vs l.31 (and elsewhere) make the use of "donor" or "source" consistent p.2 l.9 should probably be "conservation for critical quantities" p.3 l.20 the subject of "that nonlinearly couple" should not be "solution fields" (they are just exchanged in the nonlinear coupling process)
  Authors: Done
  Manuscript Diff: p.2 l.6

- Referee: p.6 l.6 unclear (if not useless) reference "Section (1)"
  Authors: Introduced a subsection 1.1 to clarify.
  Manuscript Diff: p.2 l.27 - new subsection

- Referee: p.6 l.32 the "GLL acronym" is used before definition which is given a few lines later
  Authors: Done
  Manuscript Diff: p.7 l.6

- Referee: p.7 l.29 "an in-memory" instead of "a in-memory" p.8 l.26 remove "a" before "replicated SE and MPAS" meshes
  Authors: Removed "a"
  Manuscript Diff: p.8 l.29

- Referee: p.9 Algorithm 1. - Step 1: if l can only be s or t - as in step 4: - indicate $l \in [s, t]$ also in step 1: otherwise if the formulation is generic for more than one mesh for component, the naming should be consistent. - Step 2: if you indicate $W_{ij}$ instead of $W_{st}$ you should not define i,j as a mesh pair. Later at step 20: i takes a specific meaning.
  Authors: Revised and included suggested changes
  Manuscript Diff: p.10 l.3 in Algorithm. 1

- Referee: p.12 l.1 "partitioner" instead of "repartioner" p.12 l.23 remove "is" before "results" p.12 l.26 "each" instead of "Each" p.14 Fig.7 caption: "fully covers" instead of "fully cover" p.15 l.3 "the intersection vertices [...] need" instead of "needs"
  Authors: Done
  Manuscript Diff: p.10 l.1-l.3 in Algorithm. 1

**2  Rebuttals for Reviewer 2 comments**

**2.1  General Comments**

- Referee: The main body text where references are given needs to be reformatted. The references and text are not clearly separated and make them difficult to read. There is also a reference to "Section. (1)" on Page 6, line 6, that I believe suffers from the same formatting problem?
  Authors Done.  We replaced "cite" with "citep".  Manuscript Diff: All citations/references in the manuscript have been modified due to this change.

- Referee: The outstanding questions that are not answered in the paper are (1) can the weights generated online be counted on to produce error free interpolation (conservation, monotonicity, etc) without first being reviewed and validated offline? (2) is the weights generation capability robust and reliable enough to run on different platforms and expect the same results to at least roundoff? (3) is it faster to generate weights online vs reading them in? (4) Is there some benefit to generating the weights online and then being able to reuse them as compared to regenerating them each time the model is run with regard to performance or reproducibility? It would be helpful if the paper addressed these issues if possible. These issues are partly raised in a few places in the paper, at least Page 6,

Lines 25-27 and Page 18, Lines 9-10. Some addiitonal discussion/results might be interesting.

Authors We have made changes to the manuscript in the Background, Software and Results sections to raise these questions and to address the solutions appropriately as needed. Detailed discussions have also been provided in a previous reply to the reviewer comments.

Manuscript Diff: p.7 l.25-l.31, p.20 l.21-l.22, p.25 l.20 - p.26 l.23.

- Referee: The MCT gsmap is generally compressed significantly because the information can be defined via a single start and end ID for certain kinds of decompositions. Since the MOAB mesh carries more info, I assume that compression is not possible and that the mesh consists of "n" fields of data for each gridpoint/corner/edge/etc? Is that a lot of data? Does the memory scale at all at higher resolutions and higher pe counts? I'm sure much of this is documented in MOAB papers, but it would be nice to add a sentence or two about it in this paper.

  Authors Some discussions about the mesh storage and memory requirements for serializing field DoF data on the MOAB mesh has been added. Again, a detailed discussion was provided in the previous response and we can add to it if further clarifications are needed.

  Manuscript Diff: p.9 l.7-l.12.

- Referee: In Figure 1, it looks like there is no longer a coupler. Where are the noncoupling non-mapping coupler operations (merging, atm/ocn flux, diagnostics, etc) being computed? In text, it sounds like the coupler component still exists but that the underlying MCT datatypes were swapped for MOAB datatypes, an additional set of calls were added in the component coupling layer to more fully describe the meshes, the online weights generation was added, and the online sparse matrix multiply was converted from MCT calls to MOAB calls. But then at page 17, line 12-15, it sounds like the coupling is between pairs of components excluding a coupler. It would be good if this were clarified.

Authors Quoting from our previous response: "The hub coupler still exists. We are currently duplicating the MCT calls alongside the MOAB based coupler in order to fully verify and validate both the accuracy and performance at runtime. After full validation, the MCT coupler will be completely removed from E3SM. The MOAB coupler allows the possibility for ATM to directly compute the remapping weights to project field data to OCN since the intersection will then be carried out through migration of OCN mesh to ATM pes. Hence this pair-wise coupling leads to a more distributed coupling strategy in the future. However, we do envision that there will still be a thin layer of a global coupler, even in the distributed case, to drive the subcycling, to compute merging with weighted combinations of fluxes, for validation and other diagnostics data outputs. We understand that Fig. (1) is somewhat misleading in this context and intend to make modifications to make it clearer.". Clarifications have been added to the text along these lines.
Manuscript Diff: p.9 l.33-l.34, p.12 l.6-l.11

**2.2 Technical Corrections**

- Referee: Page 6, line 20 "oas (2018)" ?
  Authors Fixed.
  Manuscript Diff: p.6 l.34.

- Referee: Page 7, line 29 fix "a in-line", should be "an in-line"
  Authors Changed "a in-memory" to "an in-memory"
  Manuscript Diff: p.8 l.29.

- Referee: Page 8, line 7 Alg. 1 -> Algorithm 1
  Authors Fixed.
  Manuscript Diff: p.9 l.18.

- Referee: Page 11, Fig 5b. It seems unlikely that the trivial decompostion would be someone's first guess for best performing decomposition with knowledge of how the coupling/ mapping work. Having said that, I'm surprised it performs as well as it does in Figure 14. There are lots of other resonable decompositions, why were Trival and Zoltan chosen to be highlighted in this paper? And why does the trivial decomposition perform so well in Figure 14.

  Authors This was another particularly interesting result from our scaling studies. The triival partitioner is not particularly the best strategy, but from an implementation stand-point, easiest to get working. However, we expected the Zoltan repartitioner to provide much better scaling and overall speedup (in terms of time) when computing the remapping weights by minimizing the source coverage mesh communication time. But, this problem is particularly tricky, since there are two parts that have to be optimized simultaneously.

  1. Migration from component to coupler requires repartitioning,
  2. computing coverage mesh requires moving source mesh elements to cover local target elements.

  So even if one partitioner is optimal for migration, it may still require moving lot of elements for coverage computation. There are ways where we could simultaneously optimize the partition for all components (source/target combinations) while at the same time taking into account the PE layouts, but this implementation is more involved, and is a work in progress at this stage.

  Manuscript Diff: p.13 l.29 - p.14 l.6.

- Referee: Page 12 line 23, remove "is" in "is results in"
  Authors Done.
  Manuscript Diff: p.14 l.8

- Referee: Page 16, line 28, bit-for-bit capability is sometimes important to achieve, certainly for identical runs, also for runs on different pe counts (sometimes with a

performance penalty via an optional flag). This sentence left me asking what the bit-for-bit capabilities are and what risks are introduced when computing online versus reusing.

Authors Agreed. If the performance penalties are not an issue, potentially exact bit-for-bit runs can be performed with the MOAB intersection. While we have not noticed any variation in the actual supermesh computation, the element sequence in the resulting supermesh will have to be re-sorted so that it is always partition agnostic. Currently, this is not strictly enforced. Additionally, any and all reductions in remapping weight computations, enforcing conservation and performing A*x, where A is the weight matrix and x is the solution vector to be projected need to be handled carefully to preserve unique order of arithmetic necessary for bit-for-bit reproducibility. Hence our statement that this is non-trivial, though necessary in the longer run as an explicit option.

Manuscript Diff: No changes.

- Referee: Page 19, Fig 9. it would be nice if the scale were not so ad-hoc and instead something more like (0,80,4). Scales like the one shown just make the figure more difficult to digest and in this case, there is no benefit to have the breaks defined as they are relative to something simpler and easier to read. Also, I'm not sure color adds anything, I think the same could be shown via a contour plot, possibly clearer and simpler still.

Authors We originally made use of contour plots but it made the appearance much less easier on the eye. The issue with presenting this data is that its a 2-D data set showing the timings for combination of source/target element combinations. We could use 3-D plots to show surfaces aligned to the computation time but drawing conclusion from such a description was not obvious. The reasoning for the chosen scale in Fig. 9 is that around 80 secs was the maximum amount of time (upper bound) for the largest source-target element combination to run ESMF in our case. The coloring provides a relative comparison with respect to

this upper bound, and shows as you have lower target elements, all libraries perform well relatively; but when there are lot more target elements, the algorithmic differences become much more obvious.

The loop over target elements is typically the sequential part in the computation. We have stressed in multiple places how we can accelerate by using OpenMP threading or task-based programming models specifically for intersection computation and also in the TempestRemap online weight matrix generation. While we don't have any results at the moment to show performance gains with such hybrid implementations, we expect to leverage the finer grain parallelism in the next iteration of the implementation refinements.
Manuscript Diff: No changes.

- Referee: Page 18, line 29 "serial runs" vs page 20, line 1 "better performance in MBTempest : : : offers avenues to incorporate task level parallelism : : :". Are these serial runs or something else? Serial in MPI but using shared memory parallelism? Is that still serial?
Authors Yes we are referring to serial in MPI but parallelism introduced either through threads or task-based programming. Since TempestRemap is a pure serial code (no MPI/OpenMP support), we had to compare serial performance on the same architectures to draw computational throughput conclusions. As mentioned above, we will include shared memory parallelism as a separate future study when we have implemented threading and/or task-based parallelism in both MOAB and perhaps TempestRemap.
Manuscript Diff: No changes.

- Referee: Page 20, lines 18-19. For the $1024^3$ test case, are weights being generated in 2d or 3d? If 3d, is this test case an order of magitude (or more) larger than the largest climate model grids? Might be worth clairfying in text.
Authors This was a full 3-D test case. Yes we used a very high-res run to showcase strong scalability of the point location algorithm in MOAB. While current production level runs still have lower DoFs compared to this study, there has been a lot of interest in doing sub-Km atmosphere resolution studies, which will push the boundaries of what is required from remapping libraries.

Manuscript Diff: No changes.

- Referee: Page 20, lines 24-26. I agree that the initialization cost is amortized for long production climate runs. But in your example, that init cost is order (hundreds) of seconds (see fig 10c). That is for a single set (pair?) of weights. In coupled climate model, there are often order (10) of these to be done. Now we're talking 1000s of seconds which starts to sound expensive in production but is certainly very expensive for short development test runs. Would it be cheaper to store the weights in a file and read them in the next time? (see general comments). Having said that, please confirm that weights are generated on each of the 1 billion gridcells ($1024^3$) in 3d. And if so, that's a lot of gridcells.

  Authors Agreed. This is a deficiency of the Kd-tree datastructure and as mentioned in the manuscript, we intend to add BVH implementations where the overall cost for the tree construction is much smaller. $O(nlog(n))$ in Kd-tree vs $O(log(n))$ in BVH-tree. The BVH implementation is a little complex and so we do not have this working correctly for large cases yet in MOAB.

  Manuscript Diff: No changes.

- Referee: Page 21, Figure 10, I am struggling to read the axes and other text on the plots

  Authors At 100% zoom in the pdf using our Adobe reader, the axes are clearly visible. However, we can try to modify the fonts slightly to get better resolution in the images.

  Manuscript Diff: Slightly zoomed image.

- Referee: Page 21, Figure 10b shows scaling to 512k pes for a problem size of $1024^3$. The final point has 2000 gridcells per process which is still relatively big.

What if you chose a problem size of $128^3$ or $256^3$ and tried to scale to 512k cores?

Authors The complexity scales as O(nlog(n)). So if we decrease the total $n$, the total work required reduces as well, which will transition more into the memory bandwidth bound regime. So while the overall time to solution may be much lower, the strong scalability may be lower as well as expected.

Manuscript Diff: No changes.

- Referee: Page 25, figure 13. Is there benefit to showing the three results (colocated plus two disjoint). The results are very similar for the three cases, at least as presented. And there is no discussion of the differences/similarities in text.

  Authors One of the key points that we wanted to highlight was the relative indifference of the algorithms to the type of PE partitioning. When we originally looked at this study, it was our belief that the fully disjoint case would perform the worst and having any level of overlap with the coupler PEs would reduce the total amount of communication for both the mesh and data. While this may be true with really strict partitioning strategies, giving the components control over how the underlying grid is partitioned results in a nearly independent rate of scalability; this is especially evident when you look at the coverage mesh computation time that shows similar trends in all three cases.

  We will add additional text in the manuscript to point out this particular conclusion from the study, which was non-intuitive at first during our experimentation.

  Manuscript Diff: p.25 l.20 - p.26 l.23

- Referee: Page 26, figure 14b. I am surprised there is so little scaling of the send/recv at NE120 and the core counts presented. I understand the claim that the absolute cost is small in all cases. I guess you are only redistributing 86k (NE120) elements, maybe that's expected then? At 128 cores, you should be transferring over 500 elements per core. Do you expect no scaling beyond that given the message size? Do you want to mention any of this in the paper?

  Authors Fig. 14 (a) shows the actual mesh migration timing. And Fig. 14(b)

shows the send/receieve scaling for the actual field data from component to the coupler PEs. After the initial setup phase through the Crystal router algorithm during the mesh migration, all communications for field data are performed point-to-point from component to coupler PEs. The lack of scaling beyond 128 cores may be related to the size of the messages here. Since we are only measuring scalability of only one field transfer here, the latency for message creation and sending (non-blocking) still dominates the actual scaling timings; we intend to follow up this study with aggregated, multi-field transfers between atm-ocn, which should show better (lower uncertainty) point-to-point communication scaling.
Manuscript Diff: p.25 l.20 - p.26 l.23, p.28 l.8-l.9

- Referee: The jump between 64 and 128 must be a machine thing, going offnode or something?
  Authors Yes this is correct. We have added additional discussions related to these results in the paper.
  Manuscript Diff: p.28 l.13 - l.16

- Referee: Please confirm that you describe which machine the tests are run on in text and it might be beneficial to include that information in the figure captions. For page 27, figure 15, maybe remind us that it's case B of Table 1 (I think that's correct) in text.
  Authors These have been mentioned in each corresponding section for serial, parallel runs e.g., 4.1, 4.2, 4.3.1. We have also modified other sections and figures where this was not clear.
  Manuscript Diff: p.25 l.19, p.27 Fig.13 caption, p.28 l.14, p.29 Fig. 14 caption, p.30 Fig. 15 caption

[revised manuscript text omitted]

---

## Editor Decision (ED1)

Dear author,

Thank you for your revised manuscript. Your manuscript answers most comments from the reviewers, but I consider that some of the reviewers' remarks are still not addressed in a fully satisfactory way. So I would like you to consider the following remarks and provide an updated manuscript addressing them, before I can consider it for publication. These remarks are grouped in "1. Your answers to Referee 1's remarks", "2. Your answers to Referee 2's remarks" and in a third section gathering additional remarks from my side.

With best regards,
Sophie Valcke

**1. Your answers to Referee 1's remarks** (the text in italic is the referee's original remark)

- *p.4 l.29 explain "consistently respecting the underlying discretization" :*
You wrote: without approximations to the component field discretizations (type$\in$ [FV,FEM]and order); please turn this mathematical sentence between the parentheses into an English one and replace FEM by FE

- *p.7 ll.30-32 in what exactly is the MBTR stack an improvement :*
Explain why this is an improvement as requested by the referee; just saying that storing the connectivity is an improvement is not a satisfactory answer if you don't explain the positive impact of doing so.

- *p.11 ll.2-3 state here (or anticipate) the rational for replacing MCT as a broker.*
Please add these arguments somewhere in the text to answer the referee's comment.

- *p.12 ll.26-29 Fig 6. is not immediate to read without some further "step to step" details in the text.*
Yes, the advancing front algorithm should be better explained. Give more details on the relation between the text and the figure. I suppose the source cells are in red and the target cells are in blue? What does the bottom figure represent?

- *pp.12-13 subsection 3.3.1 does the seed determination can be fully automated or its efficiency depend on user tuning?*
Details on this automation should be added, as part of a better explanation of the advancing front algorithm.

- *p.14 l.7 does "to all tasks" refer to tasks (or rather processes) on the source side?*
I don't understand your answer ("Description in rebuttal is detailed"); please specify "target" or "source" or "coupler" tasks.

- *p.14 l.8 "Cells [...] are sent": how are they represented?*
Again, I don't understand your answer ("Description in rebuttal is detailed"). Please add the details you describe in your answer to the referee in the text or explain where they can be found if already in the text.

- *p.14 l.10 please clarify the term "superset": a superset usually refers to inclusion of similat objects.*
Again, I don't understand your answer ("Description in rebuttal is detailed"). Please add the details you describe in your answer to the referee in the text or explain where they can be found if already in the text.

- *p.15 l.5 how expensive can be the communication of ghost intersection elements on highly distributed components?*

You wrote that you modified the text p.16, l.21. But beside the added references, I see no modification p.16, l.21.

- *p.16 l.28 "it is non-trivial to": did you find a way?*

Both referees asked for more details. Please add details in the text.

**2. Your answer to Referee 2's remarks** (the text in italic is the referee's original remark)

- *Page 6, line 20 "oas (2018)"*

For OASIS3-MCT_4.0, please cite: Valcke, S., Craig, A. and Coquart, L. (2018), OASIS3-MCT User Guide, OASIS3-MCT4.0, CECI, Université de Toulouse, CNRS, CERFACS - TR-CMGC-18-77, Toulouse, France , Technical report XXXXX

- *Page 16, line 28, bit-for-bit capability is sometimes important to achieve*

Both referees asked for more details. Please add details in the text.

- *Page 20, lines 18-19. For the 10243 test case, are weights being generated in 2d or 3d?*

Please add in the text some details about your motivation, as you detail in your reply to the referee.

- *Page 21, Figure 10, I am struggling to read the axes and other text on the plots*

You wrote that you have zoomed the graphs but I don't see any difference; please redo the plots with captions and axis readable for a printed article  (A4 format).

- *Page 25, figure 13. Is there benefit to showing the three results (colocated plus two disjoint).*

The manuscript diff you point to do  not address the referee's question. The ones that do are p.28, l8-9. But there you only mention that "the scaling of the remapping algorithm is nearly independent of the PE layout." Please discuss a bit more as, as you state, this is very counter intuitive.

- *Page 26, figure 14b. I am surprised there is so little scaling of the send/recv at NE120 and the core counts presented.*

Again, the manuscript diff you point to do not address the referee's question. Please add something on the lack of scalability above 128 cores as you detail in your reply to the referee.

- *For page 27, figure 15, maybe remind us that it's case B of Table 1 (I think that's correct) in text.*

Please do as suggested by the referee.

**3. Please consider the following additional remarks from my side** (pages and lines now refer to your gmd-2018-280-author_response-version2.pdf)

- p.2, l. 14: please consider changing "or include trivial linear transformations " by "or are not linked by any trivial linear transformations

- p.2, l. 17: change "need" by "needs"

- p.4, l.8: what does "nonlinearly" means in "Conservative remapping of nonlinearly coupled solution fields" ?

- p.4, l.17-26: Why do you split into two paragraphs "1. NC/GC" (under which you describe NC or NC/GC or GC solutions), and "2. LC/GC" (under which you have LC/GC and LC solutions)?

- P.4, l.22: What does L2 or H1 refer to?

- P.4, l.23 : define "FD" and "FV"

- P.4, l.31: what does "locate infrastructure" mean?

- P.5, l.5: please consider changing "the solution field projection between grids" with "the solution field projected on the target grid".

- P.5, l.12: define "L2"

- P.6, l.23: OASIS3-MCT is certainly not a climate application, it is a coupler

- P.7, l.3-4: You wrote "ESMF and SCRIP traditionally handle only cell-centered data that targets Finite Volume discretizations (FV to FV projections), with first-order conservation constraints" . This is not true as both ESMF and SCRIP offer also 2nd ordre conservative remapping. Please correct.

- P.7, l.6: I don't understand what "matches the areas … to the weight … " means. Please modify.

- P.7, l.30 : in "the online remapping computation uses the exact same input grids, and …", the same input grids than what?

- P.8, l.14: replace "treats" by "treat" as "which" refers to the datatypes

- P.9, l.25, please rephrase "While the MCT infrastructure only allowed for a numbering of the grid points", as MCT certainly allows for more than this!

- P.12, first paragraph and p.16, l.14-22 : Referee 1 asked you to use MPI "processes" and OpenMP "tasks" coherently through the text so please change "tasks" for "processes" in thus paragraph. Also, p.8, l.10, change "tasks" for "processes".Alos p.26, l.27, change "task" for "process".

- Figure 8, p. 17: this is almost the same than Figure 7; I don't understand what additional information does Figure 8 bring?

- P.22, l.4: please consider removing "on the BlueGene-Q machine (Mira) at ANL" as this is stated l.10.

- P.22, l.10-15: Please refer more precisely to graphs a), b) and c) in Figure 10 where appropriate and discuss the results; it looks to me that only Figure 10 c is analysed/discussed.

- P.22, last paragraph: you write "The full 3-D point location and interpolation operations provided by MOAB are comparable to the implementation in Common Remapping component used in the C-Coupler Liu et al. (2013) (Liu et al., 2013) and provide relatively much stronger scalability on larger core counts Liu et al. (2014) (Liu et al., 2014) for the remapping operation" ; how do you get to that conclusion?

- Figure 11 b: In b) how is remapped field evaluated on land point? I.e. continents seem to have a value as they are not white on the figure.

- P. 24, l.7: define CS (Cubed Sphere) the first time you write it in the text.

- Table 1 captions: Remind the resolutions and that it is for the fully colocated PE layout

- P.28, l.8, section 4.4: mention Figure 14-(b)

- P.28, l.9, section 4.4: in "is insignificant", replace "is" by "should be" as you don't have numbers to demonstrate this here

- P.28, l.12: the text in parentheses "(volume should remain similar to Fig. 14-(b))" seems contradictory to the rest of the sentence

- P.29, section 4.5, last sentence: "However, a direct comparison between these two workflows is not yet possible, but we expect the aggregated communication strategies in the crystal router algorithm Fox et al. (1989) (Fox et al., 1989) in MOAB, to provide relatively better performance at scale." Can you explain why the two workflows are not comparable? Also why do you put this sentence here in a section on "Note on Application of Weights"?

- P.30, l.14" replace "is" by "are" in "which is then consumed"

---

## Author Response (AR2)

Dear Editor,

Please find our detailed responses for the comments you provided. We have made the appropriate changes in the manuscript and also attached the latex-diff file for said changes. Please do let us know if further modifications are needed.

**1  Answers to Referee 1's remarks**

- Referee: p.4 l.29 explain "consistently respecting the underlying discretization" : You wrote: without approximations to the component field discretizations (type $\in$ [FV,FEM] and order); please turn this mathematical sentence between the parentheses into an English one and replace FEM by FE
  Authors: The suggested change has been made in the manuscript.

- Referee: p.7 ll.30-32 in what exactly is the MBTR stack an improvement : Explain why this is an improvement as requested by the referee; just saying that

storing the connectivity is an improvement is not a satisfactory answer if you don't explain the positive impact of doing so.

Authors: Storing the connectivity of the mesh elements, the real topology of component meshes along with the parallel decomposition provides MOAB the capability to perform optimizations during mesh migration, load rebalancing with Zoltan and minimize point-to-point communications based on geometric proximity of vertices (and DoFs). p.8 l.22-l.29 details the improvements of the MBTR stack already. Let us know if these explanations are still insufficient.

- Referee: p.11 ll.2-3 state here (or anticipate) the rational for replacing MCT as a broker. Please add these arguments somewhere in the text to answer the referee's comment.

  Authors: p.8 l.28 - p.9 l.6 describes the advantage of using MOAB's compact memory data structure and communication kernels over the MCT and related GSMap structures that have an $O(p)$ complexity. Moved the reference to crystal router algorithm from p.16 l.3 to p.8. The crystal router algorithm sits at the heart of MOAB aggregated communication strategies, which are far superior in terms of scalability in comparison to MCT's M-to-N communication kernels. Additional statements have been added in p.8 to address reviewer comments.

- Referee: p.12 ll.26-29 Fig 6. is not immediate to read without some further "step to step" details in the text. Yes, the advancing front algorithm should be better explained. Give more details on the relation between the text and the figure. I suppose the source cells are in red and the target cells are in blue? What does the bottom figure represent?

  Authors: Yes we have added more details here as requested. Fig 6 has been slightly modified as well and p.15 l.1-l.7 now includes some rephrasing and detailed explanation has been added to accompany the illustration. Discussions regarding determination of initial seed or starting location of the front have also been simplified.

- Referee: pp.12-13 subsection 3.3.1 does the seed determination can be fully automated or its efficiency depend on user tuning? Details on this automation should be added, as part of a better explanation of the advancing front algorithm.
  Authors: The seed or the initial cell determination proceeds through a linear search. A cell in target mesh is chosen based on local ID space, and a cell corresponding to the geometric location is identified on the source mesh. If either of these contain a hole, we need to re-seed. Hence multiple trials may be required. Page 19 l.6 has some description related to this. We have also modified description about advancing front algorithm in Page 14.

- Referee: p.14 l.7 does "to all tasks" refer to tasks (or rather processes) on the source side? I don't understand your answer ("Description in rebuttal is detailed"); please specify "target" or "source" or "coupler" tasks.
  Authors: Changes in p. 14 l.7. The "to all tasks" refers to coupler tasks since the entire remapping process, including source coverage mesh computation is performed only in coupler PEs.

- Referee: p.14 l.8 "Cells [...] are sent": how are they represented? Again, I don't understand your answer ("Description in rebuttal is detailed"). Please add the details you describe in your answer to the referee in the text or explain where they can be found if already in the text.
  Authors: Additional details have been added to p.16 l.5. Quoting original answer from rebuttal to reviewer (AC5).
  MOAB utilizes the aggregated crystal router to efficiently send small data between processes. In an all-to-all communication strategy, with log(N) steps of communication, all the processes get access to the data they need. This is used once during the setup phase to establish point-to-point communication links, which is then used later to pack and send data directly. During the field transfer from components to coupler, we pack multiple fields together in a single array to send the data to coupler, apply weight matrices on the vectors and transmit back

the fields (in a packed and aggregated fashion) to the target component. The size of such communication is on the order of DoFs in source + target.

- Referee: p.14 l.10 please clarify the term "superset": a superset usually refers to inclusion of similat objects. Again, I don't understand your answer ("Description in rebuttal is detailed"). Please add the details you describe in your answer to the referee in the text or explain where they can be found if already in the text.
  Authors: p. 16 l.9-l.10 - rephrased and removed the term superset that is causing confusion in this context.

- Referee: p.15 l.5 how expensive can be the communication of ghost intersection elements on highly distributed components? You wrote that you modified the text p.16, l.21. But beside the added references, I see no modification p.16, l.21.
  Authors: In the experiments performed for scalability of remapping algorithm in Fig. 13, the ghost communication on the coverage mesh takes less than 0.1% of total time. This is a local point-to-point communication of nearest neighbors, the cost is extremely cheap. Statements have been added in p. 17 l.7 to this effect.

- Referee: p.16 l.28 "it is non-trivial to": did you find a way? Both referees asked for more details. Please add details in the text.
  Authors: Refer to updates in p.18 l.28-l.32.

**2 Answers to Referee 2's remarks**

- Referee: Page 6, line 20 "oas (2018)" For OASIS3-MCT_4.0, please cite: Valcke, S., Craig, A. and Coquart, L. (2018), OASIS3-MCT User Guide, OASIS3-MCT4.0, CECI, Université de Toulouse, CNRS, CERFACS - TR-CMGC- 18-77, Toulouse, France , Technical report XXXXX

Authors: I could not find a report number but referenced without it. Please let us know if you have an updated reference.

- Referee: Page 16, line 28, bit-for-bit capability is sometimes important to achieve Both referees asked for more details. Please add details in the text.
  Authors: Refer to updates in p.18 l.28-l.32.

- Referee: Page 20, lines 18-19. For the 10243 test case, are weights being generated in 2d or 3d? Please add in the text some details about your motivation, as you detail in your reply to the referee.
  Authors: P. 23 l. 3 mentions that this is a 3-d test case. Additional details have been added under P. 22 l. 27, P. 23 l. 6.

- Referee: Page 21, Figure 10, I am struggling to read the axes and other text on the plots You wrote that you have zoomed the graphs but I don't see any difference; please redo the plots with captions and axis readable for a printed article (A4 format).
  Authors: Changed the layout for the sub-figures in Fig. 10 so that the axes are clearer.

- Referee: Page 25, figure 13. Is there benefit to showing the three results (colocated plus two disjoint). The manuscript diff you point to do not address the referee's question. The ones that do are p.28, l8-9. But there you only mention that "the scaling of the remapping algorithm is nearly independent of the PE layout." Please discuss a bit more as, as you state, this is very counter intuitive.
  Authors: Additional comments have been added in Page 29, near line 7.

- Referee: Page 26, figure 14b. I am surprised there is so little scaling of the send/recv at NE120 and the core counts presented. Again, the manuscript diff you point to do not address the referee's question. Please add something on the lack of scalability above 128 cores as you detail in your reply to the referee.

Authors: Page 29, Line 28. The lack of scalability has been explained and pointers to some planned future work has been discussed.

- Referee: For page 27, figure 15, maybe remind us that it's case B of Table 1 (I think that's correct) in text. Please do as suggested by the referee.
  Authors: That is correct. Done.

**3 Answers to Sophie Valcke's remarks**

- Referee: p.2, l. 14: please consider changing "or include trivial linear transformations " by "or are not linked by any trivial linear transformations
  Authors: Done.

- Referee: p.2, l. 17: change "need" by "needs"
  Authors: Done.

- Referee: p.4, l.8: what does "nonlinearly" means in "Conservative remapping of nonlinearly coupled solution fields" ? p.4, l.17-26: Why do you split into two paragraphs "1. NC/GC" (under which you describe NC or NC/GC or GC solutions), and "2. LC/GC" (under which you have LC/GC and LC solutions)?
  Authors: explanation

- Referee: P.4, l.22: What does L2 or H1 refer to?
  Authors: These are the error norms that are standard. Brief clarifications have been added

- Referee: P.4, l.23 : define "FD" and "FV"
  Authors: Expanded

- Referee: P.4, l.31: what does "locate infrastructure" mean?
  Authors: Its for location of points to compute nearest neighbor or point in element computations using bounding-boxes, Kd-tree etc. Slightly modified text to clarify.

- Referee: P.5, l.5: please consider changing "the solution field projection between grids" with "the solution field projected on the target grid".
  Authors: Clarified.

- Referee: P.5, l.12: define "L2"
  Authors: Slightly expanded. Several references already cited that provide details on these standard mathematical notations.

- Referee: P.6, l.23: OASIS3-MCT is certainly not a climate application, it is a coupler
  Authors: We do not specifically mention OASIS3-MCT as an application. It is referred to as a coupler and compared to CIME-MCT.

- Referee: P.7, l.3-4: You wrote "ESMF and SCRIP traditionally handle only cell-centered data that targets Finite Volume discretizations (FV to FV projections), with first-order conservation constraints" . This is not true as both ESMF and SCRIP offer also 2nd ordre conservative remapping. Please correct.
  Authors: Corrected.

- Referee: P.7, l.6: I don't understand what "matches the areas ... to the weight ... " means. Please modify.
  Authors: This is how the dual or the control-volume grid for a spectral element mesh is computed. The GLL node weights in the SE grid determine the area of the dual mesh element centered around the GLL point. Since the GLL points are not equi-spaced, it typically is a slow iterative procedure to converge the area of the dual grid element to exactly match the weights of the GLL quadrature node.

We added a reference (technical report from SNL) in P.6 l. 35 for the dual grid computation to make things clearer.

- Referee: P.7, l.30 : in "the online remapping computation uses the exact same input grids, and ...", the same input grids than what?
  Authors: Compared to the offline workflow. Updated text. P.7 l.20.

- Referee: P.8, l.14: replace "treats" by "treat" as "which" refers to the datatypes
  Authors: Done.

- Referee: P.9, l.25, please rephrase "While the MCT infrastructure only allowed for a numbering of the grid points", as MCT certainly allows for more than this!
  Authors: P.9 l.17. Rephrased sentence. Please verify.

- Referee: P.12, first paragraph and p.16, l.14-22 : Referee 1 asked you to use MPI "processes" and OpenMP "tasks" coherently through the text so please change "tasks" for "processes" in thus paragraph. Also, p.8, l.10, change "tasks" for "processes".Alos p.26, l.27, change "task" for "process".
  Authors: References to task have been replaced with process where relevant. Also rephrased "remapping process" to "remapping operation" to avoid confusion.

- Referee: Figure 8, p. 17: this is almost the same than Figure 7; I don't understand what additional information does Figure 8 bring?
  Authors: Figure 7 shows the actual coverage mesh that is computed from the source mesh in coupler PEs such that the local target mesh is entirely covered. Figure 8 shows the actual intersection mesh that does not have any elements beyond the target mesh, but within the local target mesh, contains the union of source and target elements. Let me know if this is still unclear or if we should add some text in relevant section to detail the differences more.

- Referee: P.22, l.4: please consider removing "on the BlueGene-Q machine (Mira) at ANL" as this is stated l.10.
  Authors: Done.

- Referee: P.22, l.10-15: Please refer more precisely to graphs a), b) and c) in Figure 10 where appropriate and discuss the results; it looks to me that only Figure 10 c is analysed/discussed.
  Authors: P.22 l.30. Description for each sub-figure has now been added.

- Referee: P.22, last paragraph: you write "The full 3-D point location and interpolation operations provided by MOAB are comparable to the implementation in Common Remapping component used in the C-Coupler Liu et al. (2013) (Liu et al., 2013) and provide relatively much stronger scalability on larger core counts Liu et al. (2014) (Liu et al., 2014) for the remapping operation" ; how do you get to that conclusion?
  Authors: We looked at performance results presented in previous references for the C-Coupler before making the statement. Specific pointers are provided below.

  - Please refer to results in Section 5.4.1 and Section 5.4.2 in C-Coupler1
  - Also refer to results in Section 5.5 in C-Coupler2

  We want to point out that the results shown with the MOAB bilinear interpolation have been shown to scale at 50% at over half a million cores on Mira, while the initialization and data transfer costs for C-Coupler as shown in the referenced papers are much higher than that of MOAB, even for much smaller resolution cases. In any case, please do let us know if we need to make the text in our manuscript a little less controversial.

- Referee: Figure 11 b: In b) how is remapped field evaluated on land point? I.e. continents seem to have a value as they are not white on the figure.

Authors: The ocean mesh does not have any elements in the land regions. So it is a mesh with holes. Hence the projection from ATM to OCN results in field data only where ocean mesh exists, and there is nothing to plot in the continent regions. This refers to Fig. 11 (b). It is possible that the confusion arises due to the solution visualization from the opposite side of the sphere that is showing through North America.

In Fig. 11 (c), the OCN mesh does not have any holes and covers the entire sphere and hence projection results in the field defined everywhere.

- Referee: P. 24, l.7: define CS (Cubed Sphere) the first time you write it in the text.
  Authors: Now CS is defined in P. 21 l.3.

- Referee: Table 1 captions: Remind the resolutions and that it is for the fully colocated PE layout
  Authors: Done.

- Referee: P.28, l.8, section 4.4: mention Figure 14-(b)
  Authors: Added reference.

- Referee: P.28, l.9, section 4.4: in "is insignificant", replace "is" by "should be" as you don't have numbers to demonstrate this here
  Authors: Done.

- Referee: P.28, l.12: the text in parentheses "(volume should remain similar to Fig. 14-(b))" seems contradictory to the rest of the sentence
  Authors: Removed.

- Referee: P.29, section 4.5, last sentence: "However, a direct comparison between these two workflows is not yet possible, but we expect the aggregated communication strategies in the crystal router algorithm Fox et al. (1989) (Fox et al., 1989) in MOAB, to provide relatively better performance at scale." Can you

explain why the two workflows are not comparable? Also why do you put this sentence here in a section on "Note on Application of Weights"?

Authors: Yes we do not use the crystal router algorithm for point-to-point communication when applying matrix weights. We have modified the description accordingly.

The two workflows are not directly comparable at the moment because for ATM (SE) to OCN (FV) projections, the offline maps generated with ESMF use the "dual" grid and online maps generated with TempestRemap use the original spectral grid, which results in very different communication graph and non-zero pattern in the weight matrix. We are still implementing a suitable infrastructure to abstract out these details so that a direct comparison can be made here. We have added some of these details to the manuscript to make the discussions clearer. Kindly let us know if the text needs to be modified further.

- Referee: P.30, l.14" replace "is" by "are" in "which is then consumed"
  Authors: Done

We request you to review the updated paper and welcome other comments that would improve the scope of the manuscript.

Best regards,

Vijay Mahadevan

[revised manuscript text omitted]

---

## Author Response (AR3)

Dear Editor,

Please find our detailed responses for the comments you provided. We have made the appropriate changes in the manuscript and also attached the latex-diff file for said changes. Please do let us know if further modifications are needed.

**1 Responses to Dr. Sophie Valcke's remarks**

- Editor: for the OASIS3-MCT_4.0 reference, please use (with the http reference if you find this appropriate) : Valcke, S., Craig, A. and Coquart, L. (2018), OASIS3-MCT User Guide, OASIS3-MCT4.0, CECI, Université de Toulouse, CNRS, CERFACS - TR-CMGC-18-77, Toulouse, France (https://cerfacs.fr/wp-content/uploads/2018/07/GLOBC-TR-oasis3mct_UserGuide4.0_30062018.pdf)

  Authors: The reference has been updated to include the URL as well.

- Editor: p.4, l10-20: It is still not clear why you you split the description into two

paragraphs "1. NC/GC" (under which you describe NC or NC/GC or GC solutions), and "2. LC/GC" (under which you have LC/GC and LC solutions)? In your reply, there is just an "explanation" written in red without further justification.

Authors: p. 4, l10-20 have been modified to make the distinctions clearer. Please let us know if the changes are sufficient. The original idea was to separate NC and GC/LC methods. LC is harder to achieve, but when every element or DoF can be conservatively computed locally, it sufficiently implies GC. A statement has been added to this effect.

• Editor: p.16-17: I still think that Fig. 8 does not bring any additional information. In particular, the "as shown in Fig. 8", L2 P17 does not seem relevant as nothing specific about the front algorithm is shown in Figure 8.

Authors: Fair point. We have now removed Fig. 8 and references to it.

• Editor: In Fig. 7 captions in "local intersection proceeds between atmosphere (Quadrangle) and ocean (Polygonal) grids", maybe add "target" before "atmosphere" and "source" before "ocean".

Authors: It is the other way around; we have modified the caption.

• Editor: Fig 11 b), yes I think that some confusion arises as the solution from the opposite site of the sphere shows through America. It would be better to have all continents in white, or at least explain why they are not all white to avoid the confusion ...

Authors: We unfortunately do not have the exact same dataset used to generate the image in Fig 11. Due to lack of time on my part, I am only updating the text to better explain why Fig. 11 (b) has transparent shading. If this still does not suffice, I will try to regenerate all three images again with a new projection run. Please do let me know your comments.

- Editor: p.8, l.31, please put the citation "Tautges and Caceres (2009)" between parentheses.

  Authors: Replaced cite with citep.

- Editor: p.9, l.4, please consider replacing "processes" by "functions" or something similar

  Authors: p.9, l.4 has been rephrased.

We request you to review the updated paper and welcome other comments that would improve the scope of the manuscript.

Best regards,

Vijay Mahadevan

[revised manuscript text omitted]